# Improved FTIR retrieval strategy for HCFC-22 (CHClF$_2$), comparisons with in situ and satellite datasets with the support of models, and determination of its long-term trend above Jungfraujoch

Maxime Prignon[1], Simon Chabrillat[2], Daniele Minganti[2], Simon O'Doherty[3], Christian Servais[2], Gabriele Stiller[4], Geoffrey C. Toon[5], Martin K. Vollmer[6], and Emmanuel Mahieu[1]

[1]Institute of Astrophysics and Geophysics, University of Liège, Liège, Belgium
[2]Royal Belgian Institute for Space Aeronomy, Brussels, Belgium
[3]Atmospheric Chemistry Research Group, School of Chemistry, University of Bristol, Bristol, UK
[4]Karlsruhe Institute of Technology, Karlsruhe, Germany
[5]Jet Propulsion Laboratory, California Institute of Technology, Pasadena, CA, USA
[6]Laboratory for Air Pollution and Environmental Technology, Empa, Swiss Federal Laboratories for Materials Science and Technology, Dübendorf, Switzerland

*Correspondence to*: Maxime Prignon (maxime.prignon@uliege.be)

**Abstract.** Hydrochlorofluorocarbons (HCFCs) are the first, but temporary, substitution products to the strong ozone depleting chloroflurocarbons (CFCs). HCFC consumption and production are currently regulated under the Montreal Protocol on Substances that Deplete the Ozone Layer and their emissions have started to stabilize or even decrease. As HCFC-22 (CHClF$_2$) is by far the most abundant HCFC in today's atmosphere, it is crucial to continue to monitor the evolution of its atmospheric concentration. In this study, we describe an improved HCFC-22 retrieval strategy from ground-based high-resolution Fourier Transform InfraRed (FTIR) solar spectra recorded at the high-altitude scientific station of Jungfraujoch (Swiss Alps, 3580 m above mean sea level). This new strategy distinguishes tropospheric and lower stratospheric partial columns. Comparisons with independent datasets (the Advanced Global Atmospheric Gases Experiment/AGAGE and the Michelson Interferometer for Passive Atmospheric Sounding/MIPAS) supported by models (the Belgian Assimilation System for Chemical ObErvation/BASCOE and the Whole Atmosphere Community Climate Model/WACCM) demonstrate the validity of our tropospheric and lower stratospheric long-term time series. A trend analysis on the datasets used here, now spanning thirty years, confirms the last decade's slowing down of the HCFC-22 growth rate. This updated retrieval strategy can be adapted for other ozone-depleting substances (ODSs) as CFC-12, for example. Measuring or retrieving ODS atmospheric concentrations is essential to scrutinise the fulfilment of the globally ratified Montreal Protocol.

## 1 Introduction

Chlorodifluoromethane (CHClF$_2$), also known as HCFC-22 (hydrochlorofluorocarbon-22), is an anthropogenic constituent of the atmosphere. It is mainly produced today for domestic and industrial refrigeration systems. As HCFC-22 is a chlorine-containing gas, it is responsible for stratospheric ozone loss and is regulated by the Montreal Protocol on Substances that Deplete the Ozone Layer. HCFC-22 has a global total atmospheric lifetime of 12 years (9.3-18 yr; SPARC, 2013) and its ozone depletion potential is 0.034 (WMO, 2014). HCFC-22 has also a significant global warming potential (1760 on a 100-yr time horizon; IPCC, 2013).

As HCFCs are the first, but temporary, substitution products to the now banned CFCs, their emissions have been on the rise. Despite the large bank of HCFC-22 remaining in refrigeration systems, HCFC-22 emissions should decrease as the Montreal Protocol and its 2007 adjustment planned a 97.5-100% reduction of the overall production of HCFC by 2030 for all countries. HCFC-22 emissions in fact increased before 2007 but have been constant since then (Montzka et al., 2009, 2015;

WMO 2014; Simmonds et al., 2017). Simmonds et al. (2017), in their recent study, determined global HCFC-22 emissions at $(360.6 \pm 58.1)$ Gg yr$^{-1}$ – representing about 79% of total HCFC emissions – and a global mean mole fraction of $(234 \pm 35)$ pmol mol$^{-1}$ for the year 2015. These results are in good agreement with previous studies of Montzka et al. (2009, 2015) and the 2014 WMO report on Ozone-Depleting Substances (ODS). The 2004-2010 trends (in % yr$^{-1}$ relative to 2007) are $3.97 \pm 0.06$, $3.52 \pm 0.08$ and $3.7 \pm 0.1$ derived from in situ, ground-based Fourier Transform InfraRed spectrometers (FTIR) and satellite measurements, respectively (WMO, 2014). Yearly global mean growth rates reached a maximum of 8.2 pmol mol$^{-1}$ yr$^{-1}$ in 2007 and decreased by 54% to 3.7 pmol mol$^{-1}$ yr$^{-}$ in 2015 (Simmonds et al., 2017). Global mean mole fractions of HCFC-22 are predicted to decrease by the year 2025 in the baseline scenario of the 2014 WMO report (see Figure 5-2 and Table 5A-2 in the report).

Nowadays, two global networks collect and share HCFC-22 in situ and flask measurements: the Advanced Global Atmospheric Gases Experiment (AGAGE) and the National Oceanic & Atmospheric Administration/Halocarbons and other Atmospheric Trace Species Group (NOAA/HATS). Alongside these 'in situ' networks, remote sensing measurements using the FTIR technique also contribute to the monitoring of HCFC-22. FTIR measurements are performed from balloon-borne (e.g., Toon et al., 1999), space-borne and ground-based platforms. The Michelson Interferometer for Passive Atmospheric Sounding (MIPAS) provided HCFC-22 satellite limb emission measurements from July 2002 to April 2012 (e.g., Chirkov et al., 2016). The Atmospheric Chemistry Experiment-Fourier Transform Spectrometer (ACE-FTS) is the only other space experiment to retrieve HCFC-22 atmospheric abundance (e.g., Nassar et al., 2006). ACE-FTS has been performing solar occultation since August 2003 although the SCISAT satellite mission was originally planned to last two years (Bernath et al., 2005). Finally, in the framework of the Network for the Detection of Atmospheric Composition Change (NDACC, De Mazière et al., 2018; http://www.ndacc.org), more than twenty ground-based stations, spanning latitudes from pole to pole, record high-resolution solar spectra with FTIR instruments. Note that only a minority of these stations currently retrieve HCFC-22 abundance.

All these measurement techniques put together enable the atmospheric scientific community to verify the fulfilment of the protocols protecting stratospheric ozone (Montreal Protocol) and reducing greenhouse gas emissions (e.g., Kyoto Protocol and Paris Agreement). The necessity of these verifications was highlighted most recently by the detection of an unexpected increase of global emissions of CFC-11 (Montzka et al., 2018).

The purpose of this study is to improve our HCFC-22 retrieval strategy such as to enhance and maximise the information content, in order to retrieve partial columns from high spectral resolution FTIR solar absorption spectra (Section 3). The resulting tropospheric and lower stratospheric updated time series are compared to independent datasets and to models (Section 4). Moreover, a trend analysis is performed in order to separate distinct HCFC-22 growth rate time periods (Section 5).

## 2 FTIR observations at the Jungfraujoch

The Jungfraujoch scientific station (JFJ), affiliated with the the NDACC network, is located on the northern margin of the Swiss Alps at 3580 m above mean sea level. Thanks to its high elevation, the station is usually under free troposphere conditions with less than 45% of air coming from the planetary boundary layer (PBL) on average (Collaud Coen et al., 2011). Consequently, the station can be considered as a "mostly remote site" (Henne et al., 2010) and experiences atmospheric background conditions over central Europe. This peculiar location also enables study of the mixing of the PBL and the free troposphere (Reimann, 2004). Indeed, the station can receive polluted air during events such as frontal passages, Föhn (Uglietti et al., 2011; Zellweger et al., 2003) or thermal uplift from the surrounding valleys (Baltensperger et al., 1997; Henne et al., 2005; Zellweger et al., 2000).

Since the mid-1980s, very high-resolution (0.003 to 0.006 cm$^{-1}$) infrared solar spectra have been regularly recorded at the Jungfraujoch station, under clear-sky conditions using wide band-pass FTIR spectrometers. In this study, spectra from the two JFJ FTIR spectrometers are exploited, i.e., a homemade instrument (1984 to 2008) and a modified Bruker IFS-120HR spectrometer (early 1990s to present). More information on these two instruments is available in Zander et al. (2008).

The spectra relevant to our study encompass the 700-1400 cm$^{-1}$ range (HgCdTe detectors), and have been recorded at two different spectral resolutions: 0.0061 cm$^{-1}$ and 0.004 cm$^{-1}$, corresponding to maximum optical path differences of 81.97 cm and 125 cm, respectively.

## 3 HCFC-22 retrieval

### 3.1 Spectroscopy

HCFC-22 presents strong absorption band systems in the infrared spectral region (Harrison, 2016): the main features are the Coriolis-coupled doublets $\nu_3$ (~1108.7 cm$^{-1}$) and $\nu_8$ (~1127.1 cm$^{-1}$) and the Q-branches $\nu_4$ (~809.3 cm$^{-1}$) and $2\nu_6$ (~829.1 cm$^{-1}$). The well-isolated but unresolved $2\nu_6$ Q-branch is one of the narrowest and most intense features of HCFC-22. It has thus been intensively used in FTIR studies from various platforms (e.g., Irion et al., 1994; Zander et al., 1994; Sherlock et al., 1997; Toon et al., 1999; Rinsland et al., 2005; Nassar et al., 2006; Chirkov et al., 2016 and Zhou et al.,

2016). As no resolved linelists are available for such relatively heavy molecules, one has to work with laboratory absorption cross-section spectra. In order to interpolate or extrapolate these cross-sections at temperatures and pressures spanning the atmospheric conditions, we use a pseudo-linelist (PLL) developed by one of us (G. C. Toon, https://mark4sun.jpl.nasa.gov/pseudo.html). The PLL used here was built by fitting the cross-sections calculated by McDaniel et al. (1991) and Varanasi et al. (1994). The main interfering species in the windows investigated to establish our

retrieval strategy (see next Section) are $H_2O$, $CO_2$ and $O_3$ and their line-by-line spectroscopic parameters are taken from HITRAN 2008 (Rothman et al., 2009).

### 3.2 Strategy

The profile inversions and column retrievals are performed with the SFIT-4 v0.9.4.4 algorithm which implements the Optimal Estimation Method (OEM) developed by Rodgers (2000). This tool corresponds to an upgrade of the SFIT-2

retrieval algorithm (Rinsland et al., 1998). We consider a 41-layer atmosphere model (above the Jungfraujoch) spanning the 3.58 km to 120 km altitude range, with thicknesses progressively increasing from ~0.65 km at the surface up to 14 km for the uppermost layer. The assumed pressure-temperature and a priori water vapour profiles are provided by the National Centers for Environmental Prediction reanalysis (NCEP; Kalnay et al., 1996) and extrapolated above 55 km by outputs from the Whole Atmosphere Community Climate Model v4 (WACCM, see Section 4.1.4). A priori profiles of HCFC-22 and all

interfering species, with the exception of water vapour, are also computed from a climatology of WACCM v4 outputs for the 1980-2020 time period. The solar line compilation supplied by Hase et al. (2006) has been assumed for non-telluric absorptions.

Three spectral ranges encompassing the $2\nu_6$ Q-branch (829 cm$^{-1}$) as well as the $\nu_4$ (809 cm$^{-1}$) and the $\nu_3$ (near 1100 cm$^{-1}$) features have been tested for the HCFC-22 retrieval at JFJ. For the latter, it appeared rapidly that the results were not

consistent. The corresponding columns were indeed excessively large, by more than 20 %, suggesting a discrepancy in intensity in the original cross-section data used to generate the PLL or a missing interference in this window. Two windows were thus defined: 808.45-809.6 cm$^{-1}$ (window 1) and 828.75-829.4 cm$^{-1}$ (window 2). Moreover, two main regularizations, OEM and a Tikhonov type L1 regularization (e.g., Steck and von Clarmann, 2001; von Clarmann et al., 2003; Sussmann et al., 2009), were tested to optimize the information content while keeping plausible retrieved profiles and minimizing the

error budget.

The optimization of the retrieval strategy has been performed using a subset of 598 spectra from the Bruker instrument covering the 1998 and 2015 years. Window 1 fitted alone gives poor results regardless of the regularization chosen. Considering only the OEM regularization (20% assumed for the diagonal terms of the a priori covariance matrix, $S_a$), more information is retrieved fitting both windows together. However, this strategy leads to the determination of

unrealistic vertical distributions with maximum concentration located in the lower stratosphere. The Tikhonov regularization leads to substantially better results (i.e., realistic profiles and more information retrieved) than the OEM regularization for any combination of windows. Regarding the choice between fitting only window 2 or both windows together, it appears that the first option enables retrieval of more information and robust vertical distributions, reducing the occurrence of profiles with negative values. The determination of the Tikhonov regularization strength (i.e., alpha parameter) has been performed

by minimizing the smoothing and the measurement errors (Steck, 2002), eventually leading to a value of 9. As the homemade instrument has a different point spacing ($6.102 \ 10^{-3} cm^{-1}$) than the Bruker one ($3.767 \ 10^{-3} cm^{-1}$), the relationship (Eq. 1) advised by Sussmann et al. (2009) is applied in order to harmonize the regularization between both instruments:

$$\frac{\alpha_1}{\alpha_2} = \frac{p_2}{p_1} , \tag{1}$$

where $\alpha_x$ are the Tikhonov strength parameters and $p_x$ are the instrument point spacings.

15       Figure 1 shows the selected window and the simulations performed by the SFIT-4 algorithm for HCFC-22 and the interfering species. These fits are typical of the spectral database in terms of signal-to-noise ratio (SNR), root-mean-square residuals, Degree Of Freedom for Signal (DOFS) and solar zenith angle. Note the good results obtained with the homemade instrument (Fig. 1, left panel) despite the weaker absorption (3%) and noisier spectra when compared to the Bruker results (Fig. 1, right panel). The final settings include an ozone profile retrieval while the $CO_2$ and $H_2O$ a priori columns are simply

scaled. Note that all the other interfering species are simulated but not adjusted, their a priori profiles are also computed from WACCM v4 outputs. Their overall contribution is less than 0.5% and is thus negligible.

### 3.3 Information content and error budget

The information content obtained from the retrieval processing has been objectively evaluated through the careful inspection of the averaging kernel matrices. The averaging kernel matrix ($A$) describes how the a priori ($x_a$) and the true ($x_t$)

vertical profiles contribute to the retrieved vertical distribution ($x_r$), according to Eq. (2).

$$x_r = x_a + A(x_t - x_a) \tag{2}$$

For the Bruker instrument, the mean column averaging kernel (Fig. 2 left frame), as well as the leading eigenvalues and eigenvectors (Fig. 2 right frame; see e.g., Barret et al., 2002), have been calculated on the basis of the 2015 retrievals subset. The mean DOFS, the trace of averaging kernel matrix or sum of eigenvalues, is 1.97, meaning that two pieces of

information can be extracted from the retrievals. Moreover, the second eigenvector (Fig. 2 right frame), with a value of 0.85, indicates that one can extract tropospheric (from surface to 11.21 km; as defined by the intersection of the eigenvector with the vertical axis) and lower stratospheric (11.21 to 30 km) columns from the retrieved total columns with 85% of information coming from the retrieval itself. Concerning the homemade instrument, based on the 1992 retrievals subset, the mean DOFS is 1.73. The eigenvectors are identical to the Bruker's but the eigenvalue for the second eigenvector is 0.68. Finally, note that

the Bruker instrument recorded lower SNR values during the year 2012 (30% smaller than 2015). Consequently, slightly lower DOFS (1.89) and second individual eigenvalue (0.8) are retrieved for this time period.

As fully described in Zhou et al. (2016), in the formalism of Rodgers (2000), the final state equation can be rewritten in order to express the total error in four components: the smoothing error, the forward model error $\varepsilon_F$, the measurement error $\varepsilon_y$ and the forward parameter error $K_b\varepsilon_b$. This last component "comes from the atmospheric (temperature,

a priori profiles, pressure, etc.), geometrical and instrumental parameters" (Zhou et al., 2016).

For the computation of the smoothing error components, we created the random part of the $S_a$ matrix by computing the relative standard deviation in HCFC-22 retrievals from MIPAS. The systematic component of the $S_a$ matrix was created

using the mean relative difference between ACE-FTS and MIPAS HCFC-22 retrievals. Regarding the off-diagonal elements, the interlayer correlation width has been set to 3 km. We assumed 5% relative systematic uncertainties for the spectroscopic parameters of HCFC-22 as assessed by G. C. Toon. We also assumed 5% for $O_3$ as reported in the HITRAN 2008 dataset (Rothman et al., 2009).

Results of the error budget are presented in Table 1. While the systematic errors are commensurate for both instruments (5.5%), the random errors differ significantly from an instrument to the other (5.6% of total random error for the homemade instrument and 2.7% for the Bruker instrument; this order will be implicit in the following). This difference is mainly coming from the random measurement error (4.7%/1%). The homemade instrument records lower SNRs compared to the Bruker instrument (85% of relative difference on coincidences after 2001). Moreover, the homemade instrument has

mostly been operated over a time period with less HCFC-22 abundance, so the HCFC-22 absorptions are thus weaker in spectra recorded by the homemade instrument, as is obvious from Fig. 1 (median absorption of 3% compared to 7% for the Bruker). For the systematic component of the error budget, HCFC-22 line intensities (5%) as well as temperature (1.8%) stand as the larger sources of uncertainty.

       Finally, we also investigated a possible effect of a misalignment of the Bruker instrument for the year 2012. We

assess the instrumental line shape random error by assuming an effective apodization of 0.9 (a value of 1 corresponding to a perfectly aligned instrument), a value consistent with our HBr cell spectra analysis. We find that such an apodization perturbation has a negligible effect on our retrieved total columns (less than 0.01%). The error is larger for partial columns: 1.2% and 0.6% for lower stratospheric and tropospheric columns, respectively.

### 3.4 Results

Monthly HCFC-22 columns retrieved with the strategy described in the previous Section are presented in Fig. 3. Due to direct local pollution caused by HCFC-22 leaks after the installation of new elevators in the JFJ scientific station (Zander et al., 2008), observations from the Bruker instrument from 1996 to the end of 2002 have been discarded. The homemade instrument was operated in the dome at the top of the station, almost outdoors, and consequently was practically not polluted (Zander et al., 2008). Retrievals with unusually poor residuals, low SNR, negative values in profiles or that did

not converge have been rejected. This corresponds to less than 8% of the whole dataset. Results include 7302 spectra spanning 1627 days and 272 different months of observations. The overlapping period (2003 to 2006) in the insert of Fig. 3 demonstrates the very good agreement between the two instruments, enabling us to treat our time series uniformly, without harmonization nor scaling

### 4 Improved HCFC-22 FTIR time series above JFJ and comparisons with independent datasets

**4.1 Description of independent datasets**

**4.1.1 In situ measurements**

       We include surface AGAGE data from the Mace Head (MHD, 55.33°N, 9.9°W, Ireland) and JFJ stations. At MHD, HCFC-22 measurements were initially carried out by a GC-MS ADS system (Gas Chromatography-Mass Spectrometry Adsorption/Desorption System; Simmonds et al., 1995) from January 1999 to December 2004. In June 2003, a GC-MS

Medusa system (Miller et al., 2008) was installed and the sampling frequency was doubled (every 2 hours). HCFC-22 measurements at the AGAGE JFJ station have been performed by a GC-MS Medusa system since August 2012. For each measurement, 2 L of sample are preconcentrated on a trap filled with HayeSep D and held at ∼−160°C. After desorption at 100°C, the compounds are separated and detected by GC-MS. HCFC-22 measurements are reported relative to the Scripps Institution of Oceanography-2005 (SIO-2005) primary calibration scale, leading to an absolute accuracy estimated at 2%

(Simmonds et al., 2017). Finally, an iterative AGAGE pollution identification statistical procedure (e.g., O'Doherty et al.,

2001; Cunnold et al., 2002) is applied to build "baseline" mole fractions time series representative of broad atmospheric regions. This method has an excellent performance compared to back trajectory methods as discussed in O'Doherty et al. (2001).

### 4.1.2 Satellite observation

HCFC-22 columns retrieved from MIPAS limb soundings (Fischer et al., 2008) are included for the comparison to our lower stratospheric time series. Envisat, the satellite carrying MIPAS, was launched on March 1 2002 and its mission ended on April 8 2012 after a loss of communication. Here we use the version V5R of MIPAS HCFC-22 retrievals described by Chirkov et al. (2016). All the spectra included are recorded in the so-called "reduced resolution mode", i.e., 0.12 cm$^{-1}$. Data are filtered as advised: only observations characterized by a visibility flag of 1 and diagonal terms of the averaging kernel matrix greater than 0.03 are kept.

### 4.1.3 BASCOE CTM

The Belgian Assimilation System for Chemical ObErvation (BASCOE) is an assimilation system for stratospheric composition (see Errera et al., 2016; Skachko et al., 2016). Its Chemical-Transport Model (CTM) is built around the Flux-Form Semi-Lagrangian kinematic transport module (FFSL; Lin and Rood, 1996) and the Kinetic Pre-Processor (KPP; Sandu and Sander, 2006). Chabrillat et al. (2018) provide an exhaustive description of the transport model and of the pre-processing of its forcing fields (i.e., meteorological reanalyses).

The chemical scheme of BASCOE was most recently described by Huijnen et al. (2016). Here we use a slightly expanded version including 65 chemical species which interact through 174 gas-phase reactions, nine heterogeneous reactions and 60 photolysis reactions.

For this study, the BASCOE CTM is driven by the European Centre for Medium-Range Weather Forecasts Interim Reanalysis (ERA-Interim; Dee et al., 2011). As in a recent age of air study (Chabrillat et al., 2018), the grid configuration relies on the native vertical grid of ERA-Interim (60 model levels up to 0.1 hPa, i.e., ~64 km) and a 2°×2.5° latitude-longitude grid. The time step is set to 30 minutes. The lower boundary condition, driving the chemical species surface concentrations throughout the simulation, are given by the "Historical Greenhouse Gas Concentrations" (HGGC) data produced by Meinshausen et al. (2017) for the Climate Model Intercomparison Project Phase 6 (CMIP6) experiments. As only few global observations are available for the starting year of our simulation (1984), we built the global atmosphere initial state from a BASCOE reanalysis of Aura MLS (Microwave Limb Sounder) for the year 2010, scaled by global constants to obtain abundances representative of 1984. These global constants were derived by computing the ratio between the global abundances of the year 1984 in the HGGC dataset and the ones of the year 2010 in the Aura MLS reanalysis.

### 4.1.4 WACCM

WACCM is a high-top Chemistry-Climate Model developed at NCAR (National Center for Atmospheric Research, Boulder, Colorado). It is a configuration of CAM (Community Atmosphere Model), the atmospheric model of the NCAR coupled Community Earth System Model. For an extensive description of the model, see Garcia et al. (2007) and Garcia et al. (2017) for WACCM, and Neale et al. (2013) for CAM.

In this study we use WACCM version 4 (WACCM4), which presents several extensions to the physical parameterization with respect to CAM version 4, such as the addition of the constituent separation velocities to the molecular diffusion, the modification of the gravity wave drag, a new long-wave and solar radiation parameterization above 65 km and a new ion and neutral chemistry model. WACCM uses a finite volume (FV) dynamical core (Lin and Rood, 1996) for the horizontal discretization. The chemistry scheme used in WACCM4 is MOZART version 3 (Kinnison et al., 2007), which contains 52 neutral species, one invariant ($N_2$), 127 neutral gas-phase reactions, 48 neutral photolysis reactions and 17

heterogeneous reactions. HCFC-22 (as well as some other HCFCs and HFCs) was not present in the default chemistry scheme and was therefore added. In this study, WACCM is run on a 1.9x2.5° horizontal grid and on a 66 vertical levels grid, with the default time step of 30 minutes.

We use here a free-running (FR-WACCM) configuration, where the dynamical fields are computed online together with the chemistry and radiation modules. This configuration differs from the specified-dynamics (SD-WACCM) option where the dynamical fields are nudged to a meteorological reanalysis (Froidevaux et al., 2018). The simulation covers the 1984-2014 period, starts from the same initial condition as the BASCOE CTM simulation and uses as lower boundary condition the same HGGC data produced by Meinshausen et al. (2017).

## 4.2 Data intercomparison methods

When comparing independent datasets, one has to account for the different vertical resolutions and most importantly, different vertical sensitivities. To do so, it is common to take as reference the instrument conveying the poorer vertical resolution and sensitivity. The other datasets are thus regridded on the reference's vertical grid using a conservative vertical regridding scheme to keep unchanged the total mass of the species (see, e.g., Sect. 3.1 in Langerock et al., 2015 or Sect. 3.1.1 in Bader et al., 2017) and then smoothed with the reference's averaging kernel matrix using the relation:

$$x_{smooth} = x_a + A\,(x_m - x_a), \tag{3}$$

where $A$ is the reference's averaging kernel, $x_a$ is the a priori profile used in the SFIT-4 retrieval and $x_m$ is the regridded observation or model profile extracted using a nearest neighbour interpolation. In this intercomparison, our product is the reference dataset. As the geometry of the observation affects the retrieved information content and as the mean geometry depends on the time of the year, we have computed seasonal averages of our individual averaging kernel matrices.

The mean relative differences given in sections 4.3 and 4.4 are reported in terms of fractional differences (FD) along with their 1σ standard deviation:

$$FD = 100\% \times \frac{1}{N} \sum_{i=1}^{N} \frac{O_{x(i)} - O_{y(i)}}{[O_{x(i)} + O_{y(i)}]/2}, \tag{4}$$

where $N$ is the number of coincidences between the compared datasets $O_x$ and $O_y$ (Strong et al., 2008), $O_x$ being our FTIR time series when compared.

## 4.3 Comparison of lower stratospheric columns

Figure 4 depicts the good agreement between the JFJ Bruker and MIPAS (at ±5° latitude around JFJ) lower stratospheric columns (from 11.21 km to 30 km for all data sources). The comparisons of this Section are performed on the datasets common time period, i.e., from 2005 to 2012.

The mean relative difference between JFJ Bruker and MIPAS is (-4.64 ± 6.09)%, which is within the range of the systematic error estimated for our measurements (5%; see Section 3.3). The BASCOE CTM time series is slightly lower than these two datasets with (9.80 ± 5.19)% mean relative difference to the JFJ Bruker time series. WACCM lower stratospheric columns are by far too small regarding the other datasets, the mean relative difference with the JFJ Bruker data is (26.4 ± 9.39)%.

As shown in Fig. 5, the four datasets are almost in perfect agreement for the lower stratospheric seasonality (note that we only use here MIPAS measurements performed at a maximum distance of 500 km from JFJ station). The lower stratospheric annual cycle is computed by subtracting the time series' calculated linear trend from the monthly mean lower stratospheric columns. Maximum values of HCFC-22 lower stratospheric columns are found in August while low values are seen in February. This seasonality is also pointed out by Chirkov et al. (2016) and is related to the seasonal cycle of the Brewer-Dobson circulation. The same version of BASCOE CTM was recently used to calculate the mean age of air from ERA-Interim reanalysis (Chabrillat et al., 2018) resulting, for this latitude band of the lower stratosphere, in an annual cycle

reaching maximum values in February-March and minimum values at the beginning of August. This result illustrates the young tropical air flooding the extra-tropical lower stratosphere during boreal summer due to the weaker mixing barrier formed by the subtropical jet (Chirkov et al., 2016).

## 4.4 Comparison of mean tropospheric mixing ratios

5          Figure 6 compares the averages of all the layers between surface to 11.21 km altitude of our HCFC-22 retrieved mixing ratio profiles (FTIR mean tropospheric mixing ratio hereafter) with AGAGE in situ time series. Note that the FTIR retrieved mixing ratios correspond to moist air values while AGAGE measurements are reported as dry air mole fractions. The difference between the two should be insignificantly small because of the very dry air conditions experienced at JFJ (Lejeune et al., 2017; Mahieu et al., 2014).

10          MHD and JFJ AGAGE baseline data agree very well for their common period (2012-2018) with relative extreme differences ranging from -1.82% to 4.70% and a mean relative difference of (0.50 ± 0.82)%, MHD recording higher concentrations. Unfiltered time series show similar results, with relative differences ranging from -4.1% to 5% and a mean relative difference of (0.51 ± 0.97)%.

          Comparisons between FTIR mean tropospheric mixing ratio and MHD daily averaged baseline measurements, over
the 1999-2018 time period, demonstrate a good consistency between the two datasets. The mean relative difference is (-1.11 ± 6.61)% with extreme values ranging from -47% to 23%. The scatter plot between FTIR mean tropospheric mixing ratio and MHD daily mean coincidences (insert of Fig. 6) shows the good correlation existing between the two datasets (coefficient of determination of the linear regression, $R^2$, is 0.77). Plotted monthly mean time series (Fig. 6) confirm the overall consistency over time between the FTIR mean tropospheric mixing ratios and the AGAGE datasets.

20          As BASCOE CTM and WACCM simulations have the same lower boundary condition, the simulated tropospheric mean mixing ratios are close to each other with a BASCOE CTM-WACCM mean relative difference of (4.18 ± 1.94)% (not shown) for all the simulation period. We also note a good agreement between BASCOE CTM and MHD, with BASCOE CTM being (3.67 ± 0.99)% lower than MHD for the 1999-2014 time period. This result is not surprising since Meinshausen et al. (2017) included AGAGE measurements to build their historical greenhouse gas concentration dataset but it gives
confidence in the proper application of the lower boundary condition in both models.

          Our tropospheric time series displays a similar seasonal cycle in phase as the lower stratospheric time series (7% peak-to-peak amplitude; not shown). However, Xiang et al. (2014) demonstrated that the HCFC-22 surface concentration annual cycle has a weak amplitude, with broad minima in summer and broad maxima in winter. Chirkov et al. (2016) also noticed a significant tropospheric cycle in their MIPAS upper tropospheric mixing ratio time series, in contrast with the in situ
situ data considered in their paper. They attributed this difference to the fact that their time series was capturing the intrusion of HCFC-22-poor stratospheric air at mid-latitudes Upper Troposphere/Lower Stratosphere (UTLS) at the time of the polar vortex breakdown (late winter/early spring). The effect of the polar vortex breakdown was also observed on nitrous oxide and chlorofluorocarbons in the UTLS by Nevison et al. (2004). This difference in annual cycle between the situ and FTIR time series could also be artificially amplified by the fact that our retrievals do not have a constant vertical sensitivity (see
total averaging kernels in Fig. 2) and present a peak of sensitivity in the UTLS, where the cycle is dominant compared to lower altitude signals (see lower left frame of Figure 15 in Chirkov et al., 2016). Note that this difference causes the layered structure (FTIR data spread around the in situ values) of the scatter plot of Fig. 6.

## 5 Trend analyses

          Trends calculated on the various datasets presented in the previous sections are discussed here. Computed trend
values are obtained from the linear term of a Fourier series (third order and half year semi period) fitted to the datasets. See

the Intra-annual model described in Gardiner et al. (2008) for more information. As significant auto-correlation is often found in geophysical time series, it is essential to take it into account when assessing the trend uncertainty. The approach described in Santer et al. (2000) is thus followed here to assess the 2σ uncertainty on the calculated trends. Along with the absolute trend values, we also compute relative trend values, taking as reference the yearly mean of the middle year of the considered period. The trend analysis is applied to all our partial column subsets, i.e., total columns, tropospheric columns (from ground to 11.21 km) and lower stratospheric columns (11.21 km to 30 km).

The overall multi-decadal 1988-2017 HCFC-22 total column trend is $(8.13 \pm 0.08) \times 10^{13}$ molec cm$^{-2}$. The trends for the tropospheric and lower stratospheric columns, computed over the same time period, are $(5.1 \pm 0.1) \times 10^{13}$ molec cm$^{-2}$ and $(2.99 \pm 0.05) \times 10^{13}$ molec cm$^{-2}$, respectively.

The decadal trends calculated on total columns (Table 2) show relatively high values for the late 1980s and early 1990s, i.e., $(8.52 \pm 0.57) \times 10^{13}$ molec cm$^{-2}$ for 1988-1997. The uncertainty is also greater due to the poorer sampling during this period. The models show significantly lower trends for the same time period. Temporarily lower trend values are then observed, $(7.09 \pm 0.37) \times 10^{13}$ molec cm$^{-2}$ for 1996-2005, before reaching again the same values as during the late 1980s, i.e., $(8.6 \pm 0.28) \times 10^{13}$ molec cm$^{-2}$ for 2005-2014. This evolution is well captured by the models, although WACCM is showing systematically lower absolute trend values. Finally, the HCFC-22 accumulation rate seems to slow down in the most recent time period (2008-2017). The models cannot support this later observation as the simulations end in 2014 (see Section 4.1).

Trends calculated on our tropospheric mean mixing ratio time series agree substantially well to trends calculated using AGAGE data (Table 3). The results show, as for total columns trends, the decrease of trends during the last decade (trends ~19% lower over 2008-2017 than over 1999-2008). For the overlapping period (2012-2017), the trends are also in good agreement within the uncertainties.

Concerning the lower stratospheric time series, we also include partial columns (from 11.21 km to 30 km at ±5° latitude around JFJ) from ACE-FTS (v3.6), pre-processed following the averaging kernel smoothing method described in Section 4.2. Note that MIPAS data are available for the 2005-2012 period, a bit shorter than the 10 years period used for the other lower stratospheric datasets (i.e., 2005-2014). Table 4 reports an excellent agreement within the uncertainties between the observational dataset trends, with a $(2.99 \pm 0.05) \times 10^{13}$ molec cm$^{-2}$ calculated trend for the JFJ time series.

The WACCM lower stratospheric absolute trend is too low by more than 20% compared to observations and BASCOE CTM. Since the WACCM and BASCOE CTM simulations started from the same initial condition in 1984 and use the same lower boundary condition, this bias may be due to the unconstrained dynamical fields in WACCM, in contrast with the ERA-Interim dynamical fields used by the BASCOE CTM. Consequently, the lower absolute trend in WACCM results in a significant underestimation after 2002 (see Fig. 4 and Section 4.3). The corresponding relative trends, on the other hand, do not significantly differ between observations and models.

**6 Conclusion**

Using the narrow and well isolated $2\nu_6$ Q-branch of HCFC-22, we have established an improved strategy to retrieve HCFC-22 abundances from ground-based high-resolution FTIR solar spectra. Our new approach, using a Tikhonov regularization, retrieves enough information to distinguish two independent pieces of information representative of the troposphere and the lower stratosphere. We retrieve total columns with $(66 \pm 6)\%$ (2σ) of tropospheric contribution. The main potential improvement that could be brought to our retrieval strategy would be to build a new pseudo-linelist from recently determined cross-sections (Harrison, 2016). The systematic uncertainty could be this way minimized and, moreover, the $\nu_3$ feature (near 1100 cm$^{-1}$) could be investigated again.

The comparison with independent datasets confirms the consistency and validity of these new time series. We have compared mean tropospheric mixing ratios, obtained from our retrievals, to AGAGE measurements performed at Mace Head

(Ireland) and Jungfraujoch. Despite the larger variability found in the FTIR data, our mean tropospheric mixing ratios compare very well to the in situ time series. Retrieved lower stratospheric columns are also in excellent agreement with MIPAS observations. Relative differences between MIPAS and FTIR retrievals are indeed within the systematic uncertainty assessed on our time series.

5         BASCOE CTM and WACCM outputs have been included in this study to support our comparisons. Analysis of tropospheric time series showed that the lower boundary condition chosen (Meinshausen et al., 2017) drives well the models' lower boundary through the simulation time period. WACCM lower stratospheric columns are nevertheless significantly too small compared to the observations and BASCOE CTM.

        Bias aside, we showed that all the stratospheric datasets used here depict the same seasonality in the lower
stratosphere with high values in late summer (August) and low values in late winter/spring (February and March). This seasonality was also identified by Chirkov et al. (2016) using MIPAS global limb soundings and it is neatly anti-correlated with the mean age of air derived from a BASCOE CTM simulation driven by ERA-Interim. Zhou et al. (2016) also noted this seasonality from ground-based FTIR measurements at 21°S (Reunion Island) despite the limited vertical resolution of their ground-based FTIR data (i.e., ~1 DOFS).

15         We also performed a trend analysis on the datasets used for the comparisons. The results are in good agreement for the datasets for the selected time frames. Total column and mean tropospheric mixing ratio trend analysis shows that HCFC-22 growth rates have changed significantly during the past 30 years. We confirm the decreasing of HCFC-22 growth rate during the last decade as observed by recent in situ studies (Montzka et al., 2015; Simmonds et al., 2017)

        The fact that HCFC-22 emissions are constant since 2007 and therefore that HCFC-22 growth rate has ceased to
exhibit a continuous increase over the last decade, as highlighted in this paper and other works, seems to promise a fulfilment of the Montreal Protocol and its amendments for the years to come. Recent HCFC trends even suggest that the 2013 cap on global production (Montreal Protocol) has been respected well in advance (Montzka et al., 2015; Mahieu et al., 2017). Finally, this improvement in retrieval strategy, leading to partial columns determination, could be applied to other source gases essential for monitoring chlorine in the atmosphere (e.g., CFC-12 which presents relatively narrow features).


*Code and data availability.* The FTIR data are available upon request (maxime.prignon@uliege.be). The code and outputs of BASCOE CTM are also available upon request. AGAGE date used in this work are available at http://agage.mit.edu/data/agage-data, MIPAS data at https://www.imk-asf.kit.edu/english/308.php and ACE data at https://databace.scisat.ca/level2/ .


*Author contributions.* SC and DM provided advice on the interpretation of the model data. DM performed the FR-WACCM run and SC the BASCOE age of air simulation. SOD and MKV were involved in the GCMS measurements and provided advice on the use of the AGAGE data, GS on the use of the MIPAS data. MKV furthermore provided useful information on in situ measurement techniques and helped during the discussion process. GCT computed and supplied the HCFC-22 pseudo
linelist. CS is taking care of the FTIR instrumentation maintenance and development at JFJ. MP performed the FTIR retrievals and the BASCOE CTM runs. He conducted the comparisons and trend analyses. MP wrote the manuscript and included the comments, suggestions and additions received from all the co-authors. EM supervised the work from beginning to end.

*Competing interests.* The authors declare that they have no conflict of interest.

*Acknowledgments.* Maxime Prignon and Daniele Minganti are financially supported by the F.R.S. – FNRS (Brussels) through the research project ACCROSS (PDR.T.0040.16). The University of Liège contribution was further supported by

the F.R.S. – FNRS (Grant CDR.J.0147.18), the GAW-CH program of MeteoSwiss and the Fédération Wallonie-Bruxelles. Emmanuel Mahieu is a Research Associate with the F.R.S. – FNRS. We are grateful to the International Foundation High Altitude Research Stations Jungfraujoch and Gornergrat (HFSJG, Bern) for supporting the facilities needed to perform the FTIR and in situ observations at the Jungfraujoch. We gratefully acknowledge the vital contributions of all the Belgian

5      colleagues who have performed the FTIR observations used here. We further thank Y. Christophe and Q. Errera (BIRA-IASB, Brussels, Belgium) for their contributions to the development of BASCOE. AGAGE measurements at Mace Head are supported by grants from the Department of the Business, Energy & Industrial Strategy (BEIS, UK), and the National Aeronautics and Space Administration (NASA; USA) to the University of Bristol; and AGAGE measurements at Jungfraujoch by grants from the Swiss Federal Office for the Environment (FOEN) to Empa (Halclim / CLIMGAS-CH

10    project). Part of this work was conducted at the Jet Propulsion Laboratory, California Institute of Technology, under contract with NASA. The Atmospheric Chemistry Experiment (ACE), also known as SCISAT, is a Canadian-led mission supported primarily by the Canadian Space Agency. We thank the reanalysis centres European Centre for Medium-Range Weather (ECMWF) and NOAA NCEP for providing their products. WACCM is a component of NCAR's CESM, which is supported by the NSF and the Office of Science of the US Department of Energy.

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

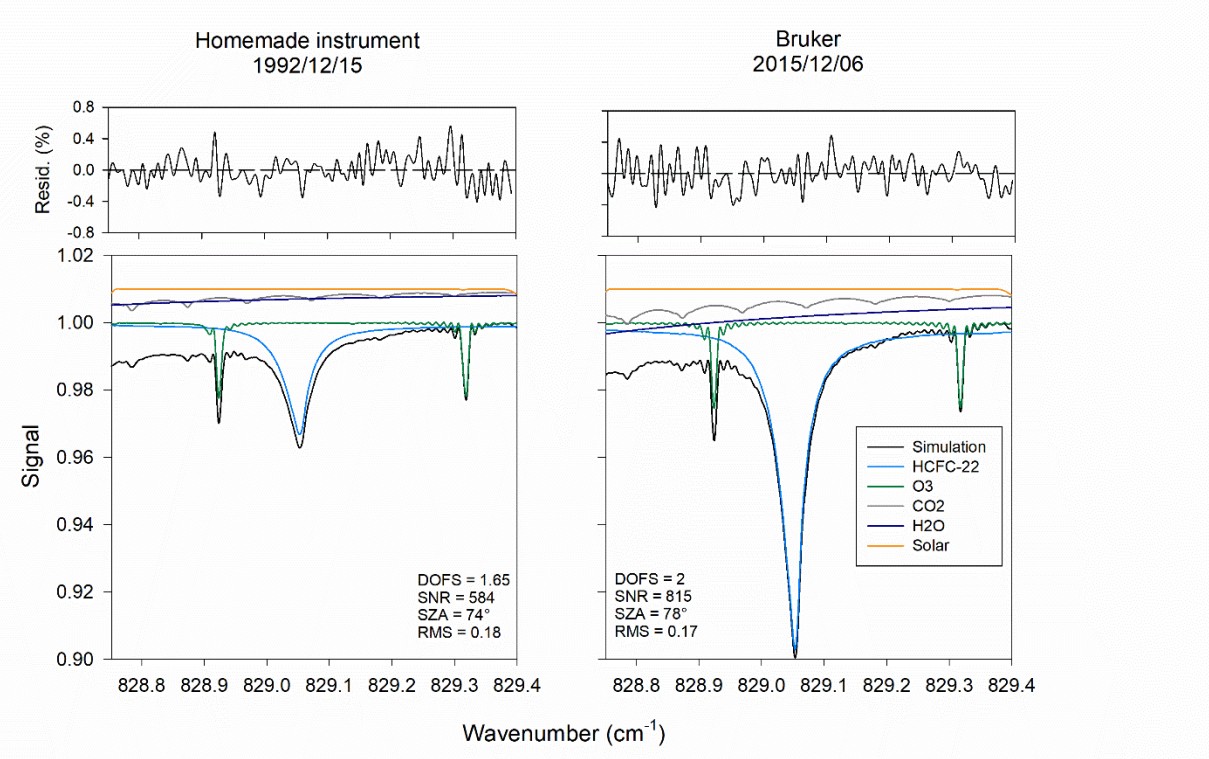

**Figure 1: Simulations of the 828.5−829.4 cm⁻¹ spectral window from spectra recorded by the homemade (1992/12/15) and Bruker(2015/12/06) FTIR instruments at Jungfraujoch. $CO_2$, $H_2O$ and solar spectra are offset vertically for clarity. Upper frames display relative residuals (%) from the fits to the spectra. These fits are typical of the spectra database in terms of SNR, root mean square error, DOFS and solar zenith angle, regarding their respective year. Note that the lower frame vertical scales correspond to only 10% of the signal amplitude.**

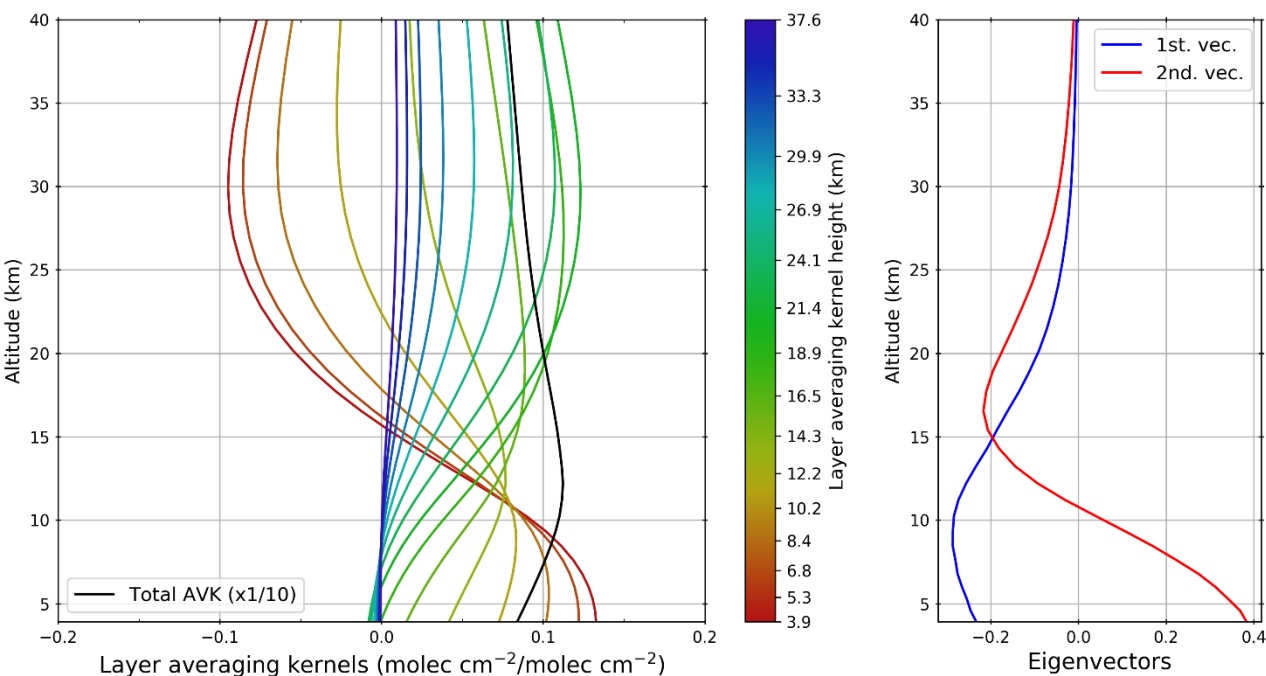

**Figure 2: Mean layer averaging kernels (left) normalized for partial columns (molec cm⁻²/molec cm⁻²) and eigenvectors (right) characterizing the FTIR retrievals of HCFC-22 above Jungfraujoch from spectra recorded in 2015 by the Bruker instrument. The ticks on the colour bar are the individual layer averaging kernels represented in the plot. The first eigenvector has a value of 1 for both instruments, and their second eigenvector has a value of 0.68 and 0.85 for the homemade and Bruker instruments, respectively.**

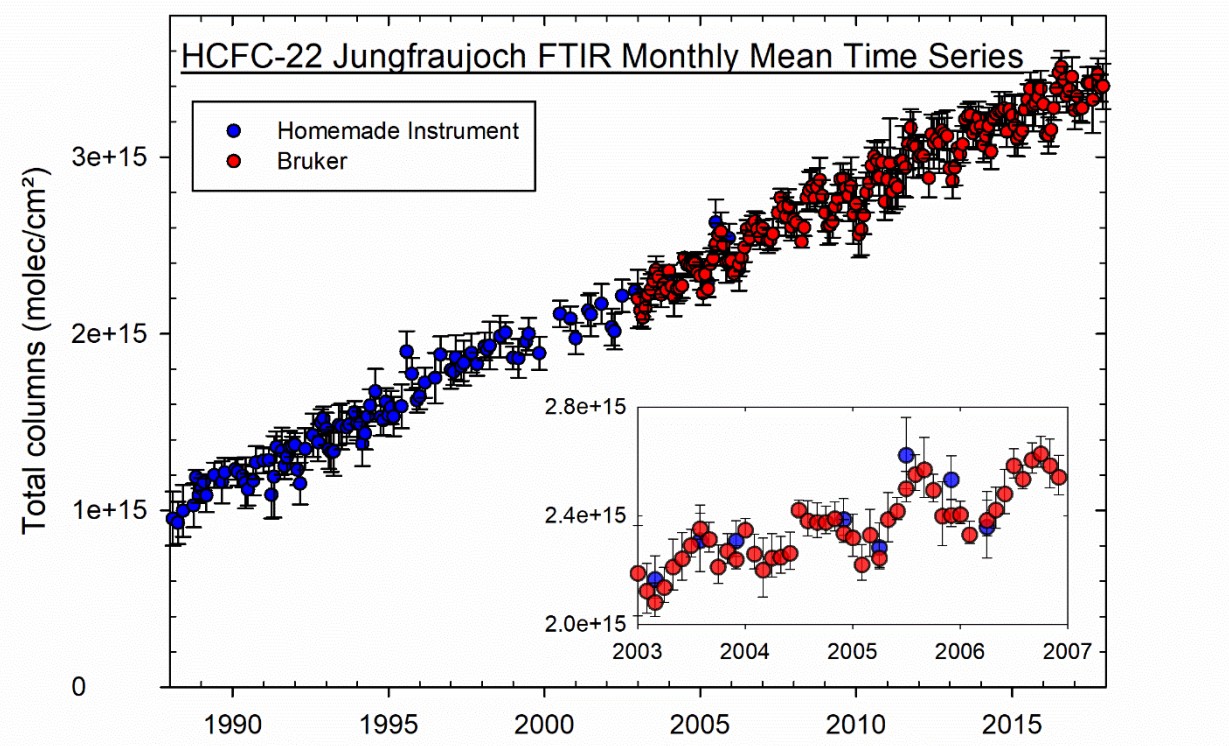

**Figure 3: FTIR monthly time series of HCFC-22 total columns above Jungfraujoch derived from spectra recorded by the homemade FTIR (blue) as well as by the Bruker IFS-120HR (red). Vertical bars are the standard deviations around the monthly means. Due to pollution events starting in 1996 and mainly influencing the Bruker instrument, observations retrieved from the Bruker spectra are discarded before 2003. Note the excellent agreement between the two instruments (insert frame).**

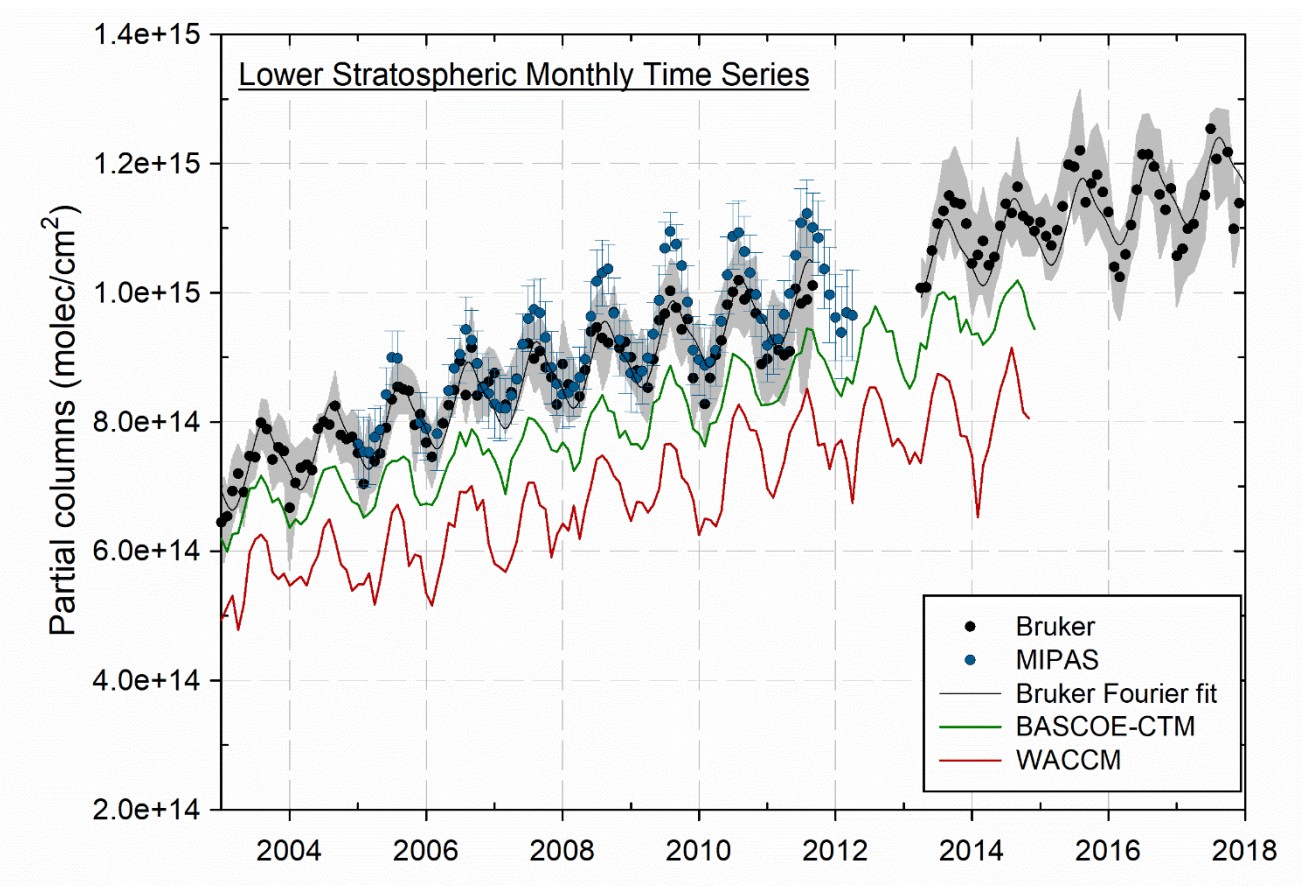

**Figure 4: Time series of lower stratospheric partial columns (11.21 to 30 km, as defined by the retrieval information content) above Jungfraujoch (MIPAS at ±5° latitude around JFJ). Grey shade and blue vertical bars depict the standard deviation around the FTIR and MIPAS monthly means, respectively. A Fourier series fitted to the Bruker time series (black curve) is also represented (see Section 5). FTIR partial columns from 2011.5 to 2013.5 are not displayed because of the lower quality retrievals observed during this time period (see Section 3.3).**

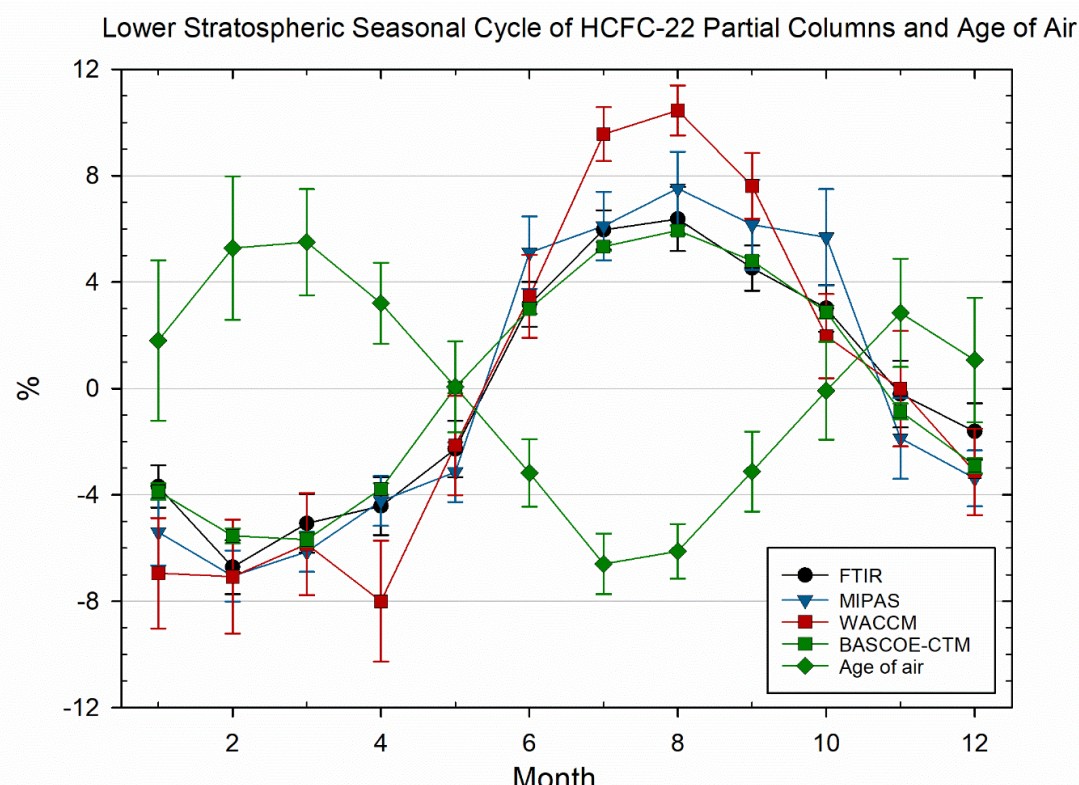

**Figure 5: Seasonal cycle of HCFC-22 columns (see Section 4.3 for method) in the lower stratosphere (11.21 to 30 km) based on measurements and model outputs (2005-2012). MIPAS measurements are at a maximum distance of 500 km from JFJ station. Vertical bars depict the 2σ standard error of the means. Age of air simulation is performed by BASCOE-CTM from ERA-Interim reanalysis. The peak-to-peak amplitude of the age of air cycle is 0.37 year and the mean age of air is 2.96 year.**

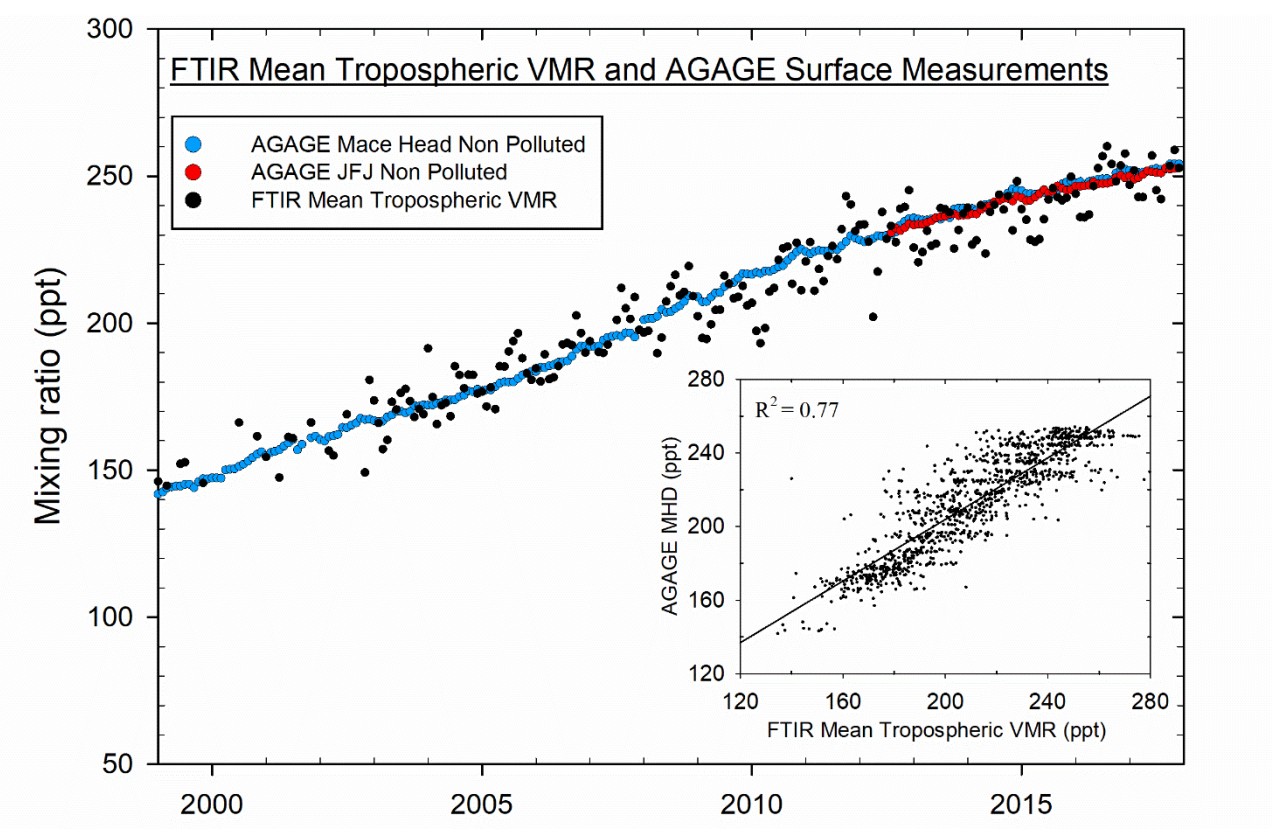

**Figure 6: Tropospheric monthly time series at Jungfraujoch. FTIR time series (black) is constructed by taking the average of all the layers below 11.21 km, the altitude limit objectively defined by the retrieval information content. AGAGE in situ time series from Mace Head (light blue) and JFJ (red) are baseline measurements (see Section 4.1.1). Daily coincidences between Mace Head and FTIR are depicted in the lower right scatter plot. The coefficient of determination of the linear regression, R², is 0.77 (R = 0.88).**

**Table 1. Mean relative errors (%) for both instruments (homemade and Bruker) affecting the total column retrievals of HCFC-22 of the years 1992 (homemade instrument) and 2015 (Bruker). See notes and text (3.3) for more information on values assumed and methods.**

| Error type (%) | Homemade | Bruker | Notes |
|---|---|---|---|
| **Random** | | | |
| – Measurement | 4.7 | 1 | OEM formalism |
| – Temperature | 2 | | |
| – SZA | 1.15 | | Assuming 0.1° for solar pointing |
| – Zero level offset | 1 | | |
| – Interfering species | 0.9 | 0.3 | |
| – Smoothing | 0.8 | 0.3 | OEM formalism |
| – Retrieval parameters | 0.5 | 0.1 | OEM formalism |
| – **Total** | **5.6** | **2.7** | |
| **Systematic** | | | |
| – HCFC-22 line intensity | 5 | | Assuming 5% from pseudo-linelist |
| – Temperature | 1.85 | | |
| – Zero level offset | 1 | | |
| – HCFC-22 air-broadening of line width | 0.9 | 0.45 | |
| – Retrieved interfering species (i.e., $O_3$) line intensity | 0.43 | 0.13 | Assuming HITRAN 2008 uncertainty |
| – **Total** | **5.5** | | |

**Table 2: HCFC-22 total columns trends over JFJ. The uncertainties are given for the 2σ level following the Santer et al. (2000) approach. Models trends are underlined when not significantly different from observations. Relative trends (% yr$^{-1}$) are given with respect to the yearly mean of the middle year of the considered period.**

| Total columns trends (10$^{+13}$molec cm$^{-2}$ yr$^{-1}$) | 1988 – 2017 | 1988 – 1997 | 1996 – 2005 | 2005 – 2014 | 2008 – 2017 |
|---|---|---|---|---|---|
| FTIR | 8.13 ± 0.08 (3.75 ± 0.04)% | 8.52 ± 0.57 (5.88 ± 0.39)% | 7.09 ± 0.37 (3.41 ± 0.18)% | 8.6 ± 0.28 (3.03 ± 0.1)% | 7.98 ± 0.29 (2.57 ± 0.09)% |
| BASCOE | | 7.21 ± 0.1 (5.3 ± 0.07)% | 6.94 ± 0.15 (3.57 ± 0.08)% | 8.98 ± 0.2 (3.35 ± 0.08)% | – |
| WACCM | | 6.63 ± 0.13 (5.16 ± 0.1)% | 6.3 ± 0.12 (3.41 ± 0.07)% | 8.53 ± 0.2 (3.36 ± 0.08)% | – |

**Table 3: HCFC-22 tropospheric trends over JFJ. The uncertainties are given for the 2σ level following the Santer et al. (2000) approach. Trends are underlined when not significantly different from FTIR trends. Relative trends (% yr$^{-1}$) are given with respect to the yearly mean of the middle year of the considered period. [a] Time frame enlarged in order to encompass the 2012 low quality measurements period (see 3.3).**

| Tropospheric trends (ppt/year) | 1988 – 2017 (10$^{+13}$ molec cm$^{-2}$) | 1999 - 2008 | 2008 - 2017 | 2012-2017 |
|---|---|---|---|---|
| FTIR Mean Tropospheric mixing ratio | 5.1 ± 0.06 (3.41 ± 0.07)% | 6.54 ± 0.35 (3.72 ± 0.2)% | 5.31 ± 0.42 (2.29 ± 0.18)% | 4.04 ± 0.73 (1.72 ± 0.31)% (2011-2017)[a] |
| AGAGE Mace Head | | 6.39 ± 0.1 (3.66 ± 0.06)% | 5.36 ± 0.12 (2.27 ± 0.05)% | 4.2 ± 0.08 (1.71 ± 0.03)% |
| AGAGE JFJ | | - | - | 4.05 ± 0.12 (1.65 ± 0.04)% |

**Table 4.** HCFC-22 lower stratospheric trends over JFJ. The uncertainties are given for the 2σ level following the Santer et al. (2000) approach. Relative trends (% yr$^{-1}$) are given with respect to the yearly mean of the middle year of the considered period. Trends are underlined when not significantly different from the FTIR trends.

| | FTIR JFJ | FTIR JFJ | MIPAS | ACE-FTS | BASCOE CTM | WACCM |
|---|---|---|---|---|---|---|
| | 1988-2017 | 2005-2014 | 2005-2012 | 2005-2014 | 2005-2014 | 2005-2014 |
| Lower stratospheric trends ($10^{+13}$molec cm$^{-2}$ yr$^{-1}$) | 2.99 ± 0.05 (4.11 ± 0.07)% | 3.11 ± 0.19 (3.29 ± 0.2)% | 3.21 ± 0.11 (3.32 ± 0.11)% | 2.96 ± 0.39 (3.17 ± 0.42)% | 2.94 ± 0.11 (3.5 ± 0.13)% | 2.53 ± 0.16 (3.47 ± 0.22)% |

