# Peer review of "Improved FTIR retrieval strategy for HCFC-22 (CHClF2), comparisons with in situ and satellite datasets with the support of models, and determination of its long-term trend above Jungfraujoch"

_Atmospheric Chemistry and Physics, 2019_

## Referee Comment (RC1) · Anonymous Referee #1 · 18 Apr 2019

<General Comments>

This paper describes an improved HCFC-22 retrieval strategy from ground-based FTIR solar spectra at Jungfraujoch. They showed the possibility to distinguish the tropospheric and lower stratospheric partial columns from the FTIR spectra and compared their results with independent datasets (AGAGE and MIPAS) and models (BASCOE CTM and WACCM). However, there are some issues that should be clarified before this paper is published in ACP, which are described in the comments below.

[Figure]

<Major Comments>

1) I have a concern on comparison between AGAGE (MHD and JFJ) data and FTIR mean tropospheric mixing ratio shown in Section 4.3. First of all, the way to calculate mean tropospheric mixing ratio from FTIR data is not described in detail. I think that SFIT-4 retrieval of FTIR spectra gives total column and vertical profiles with averaging kernel information. How the authors derive mean tropospheric mixing ratio from that information? Do they divide tropospheric HCFC-22 column between station altitude and 11.21 km by the amount of air molecule numbers at the same altitude range? Please explain in the text.

2) Annual variations are seen in both derived total columns (Fig. 3) and tropospheric mean VMR (Fig. 4) in FTIR data, both having peaks in summer to fall. Such annual variations are not seen in AGAGE MHD nor JFJ data. However, there are no explanations nor discussion on the cause of the derived annual variation. I wonder the derived annual variation may come from two reasons: a) The nature of FTIR measurement principle, i.e. measuring column amount above the observational station. The column amount might be affected by the height of tropopause height, which is higher in summer. b) The higher emission of HCFC-22 from the regional summertime use of air-conditioner, as is pointed out by Xiang et al. (2014). Please discuss more about the cause of the retrieved annual variation in FTIR data which are not seen it AGAGE data.

3) The scatter plot in Fig. 4 looks somewhat strange. We see many dots which are horizontally aligned. For example, there are several points for MHD value of ~145, but the next group jumps to >160. However, the actual trend of MHD values look more continuous. Please check if something wrong appeared or not to create this scatter plot.

4) In Section 4.4 (P.8, L.19), the authors claim that they do not show amplitude and phase of the seasonal cycle of tropospheric column series. However, as I mentioned

in the above comment, differences in tropospheric annual variations are seen between FTIR retrieval and AGAGE data. I think they should show the figure which shows amplitude and phase of seasonal cycle of tropospheric columns as well, and discuss on the cause of such variation in more detail.

<Minor Comments/Typos>

1) Throughout the main text: A new paragraph should be indented.

2) Abstract: Even in the Abstract, abbreviation for the following words should be given separately: AGAGE, MIPAS, BASCOE, and WACCM.

3) P.1, L.33: The global warming potential of HCFC-22 should be 1810 (IPCC AR4) or 1780 (WMO O3 Assessment 2018), not 1760.

4) P.6, L.9: Abbreviation for BASCOE should be given.

5) P.7, L.25: data compare very well –> data agree very well

6) P.8, L.2: Is the same altitude range (11.21-30 km) used to create the MIPAS lower stratospheric column? Please clarify.

7) Figure 6: What is the value of the age of air? I think the right hand side axis to show the age of the air is missing.

8) P.9, L.4: mixing ratio series compare –> mixing ratio series agree

---

## Referee Comment (RC2) · Anonymous Referee #2 · 28 Apr 2019

General Comments

This manuscript presents an improved retrieval strategy for deriving HCFC-22 from ground-based solar absorption spectra of the $2\nu6$ Q-branch recorded by two FTIR spectrometers at the NDACC Jungfraujoch station. The retrieval uses the SFIT-4 code with Tikhonov regularization to obtain independent tropospheric and lower stratospheric partial columns of HCFC-22, as well as total columns. The resulting time series extends from 1988 to 2017 and is used to derive trends in tropospheric, lower stratospheric, and total columns over the 30-year record and to examine differences between

individual decades in the tropospheric and total column trends. The HCFC-22 growth rate in the troposphere is shown to have slowed in the last decade. The FTIR mean tropospheric mixing ratios agree well with AGAGE in situ data and the lower stratospheric columns agree well with MIPAS satellite data. The FTIR measurements are also compared with the BASCOE CTM and WACCM models, showing that WACCM lower stratospheric columns are biased low relative to the FTIR and BASCOE. All of the lower stratospheric datasets exhibit a seasonal cycle that is anti-correlated with the mean age of air derived from a BASCOE-CTM simulation.

The results of this study are of importance in monitoring the effectiveness of the Montreal Protocol and its amendments, particularly given the length and vertical sensitivity of the FTIR data record at Jungfraujoch. The FTIR retrieval strategy for HCFC-22 could be extended to other NDACC stations and could be adapted and applied to other chlorine-containing gases.

The manuscript provides a clear and straight-forward description of the work. I recommend publication in ACP after the minor comments below are addressed.

Specific Comments

Page 5, lines 36-39 – Is this the filtering referred to on page 7, line 26? Explain on page 7 the difference between the filtered and non-filtered in situ time series. This is the first time filtering is mentioned.

Page 6, Section 4.1.2 – ACE-FTS HCFC-22 measurements are mentioned in the Introduction, so why aren't they included in the comparisons with the FTIR retrievals? Briefly explain why in this section. ACE data are also mentioned in the Data Availability section – is this because they was used in determining the systematic component of the $S_a$ matrix?

Page 6, lines 19-24 – Clarify what the lower boundary conditions are for – all trace gases in BASCOE? "only [a] few global observations are available . . ." – a few observations of what? Make clear which lower boundary conditions are derived from MLS and which from HGGC.

Page 7, Section 4.3 – Some additional explanation should be provided regarding the comparison between the FTIR mean tropospheric mixing ratio and the in situ surface mixing ratio. Why compare the FTIR tropospheric mean mixing ratio rather than the FTIR surface value or lower tropospheric mean (I assume due to the information content)? How representative is the mean tropospheric mixing ratio of the surface mixing ratio? What error does this introduce? Figure 4 suggests that there are differences, since the FTIR mean values exhibit more seasonality than the in situ values.

Page 8, lines 35-37 – Add these 1988-2017 total column trends to Tables 2-4.

Page 10, lines 9-10 – Explain more explicitly how the improved retrieval strategy developed for HCFC-22 is transferable to other gases.

Page 18, Figure 4 – Why does the AGAGE vs. FTIR scatter plot have a layered structure, with a relatively constant AGAGE value for a range of FTIR values?

Page 21-22, Tables 2-4 – Why are different time periods used for calculating the trends in the total columns, tropospheric columns, and lower stratospheric columns? This makes it difficult to directly compare them. The choices should be more clearly explained in the text.

Technical Corrections

The manuscript should be reviewed carefully for grammatical and typographical errors. For example, there are many missing commas and hyphens, unnecessary or missing "s" on words, and other errors. Some are identified below, but this list is not exhaustive.

Abstract, Page 1, line 15 – HCFCs

Abstract, Page 1, line 15 – first, but temporary,

Abstract, Page 1, line 15 – change "as" to "for"

Abstract, Page 1, line 15 – ozone-depleting

Abstract, Page 1, line 16 – CFCs

Abstract, Page 1, line 17 – Layer,

Abstract, Page 1, line 20 – Fourier Transform infrared is used here, but Fourier Transform InfraRed on page 2, para 1

Abstract, Page 1, line 20 – high-altitude

Abstract, Page 1, line 24 – delete miscellaneous

Abstract, Page 1, line 25 – ozone-depleting substances (ODSs)

Page 1, line 30 – chlorine-containing

Page 1, line 35 – HCFCs

Page 1, line 35 – change "as" to "for"

Page 1, line 35 – CFCs

Page 1, line 40 – total HCFC emissions

Page 2, line 4 – satellite

Page 2, line 13 – space-borne and ground-based

Page 2, line 14 – delete "has"

Page 2, line 15 – ACE-FTS has been performing solar occultation measurements since

Page 2, line 16 – although

Page 2, line 19 – high-resolution

Page 2, line 19 – stations currently retrieve HCFC-22 abundance.

Page 2, line 22 – could also mention the Paris Agreement here

Page 2, line 23 – by the detection of an unexpected

Page 2, lines 26 and 28 – Here and throughout the manuscript, "section" should be changed to "Section" or "Sect." following ACP formatting guidelines. Do a search and replace.

Page 2, line 31 – affiliated with the

Page 2, line 32 – conditions

Page 2, line 34 – experiences

Page 2, line 35 – This particular location also enables study of the

Page 2, line 36 – not clear what Fohn refers to – delete?

Page 2, line 40 – study,

Page 3, line 2 – detectors), and have been recorded at two spectral

Page 3, line 9 – well-isolated

Page 3, line 23 – National Centers

Page 3, line 24 – and extrapolated above

Page 3, line 34 – were tested to optimize

Page 3, line 37 – fitted alone gives poor

Page 4, line 3 – enables retrieval of more

Page 4, line 11 – spectral

Page 4, line 11 – signal-to-noise ratio (SNR), root-mean-square

Page 4, lines 19-21 – Here and throughout the manuscript, all vectors and matrices should be in bold font.

Page 4, line 24 – DOFS, the trace of the averaging

Page 4, line 25 – change "expresses" to "indicates"

Page 4, line 33 – four components

Page 4, line 34 – bold font for the matrix term

Page 4, line 38 – off-diagonal

Page 4, line 39 – set to 3 km

Page 4, line 39 – 5% relative

Page 5, line 6 – abundance, so the

Page 5, line 7 – as is obvious

Page 5, line 19 – outdoors

Page 5, line 32 – have been performed . . . since

Page 6, line 14 – nine heterogeneous

Page 6, line 17 – age-of-air study

Page 6, line 19 – Is LBC really needed as an acronym? Could just use "lower boundary condition" throughout for better clarity.

Page 6, line 22 – for the year

Page 6, line 23 – HCFC-22, the

Page 6, line 34 – (Kinnison et al., 2007)

Page 6, line 37 – horizontal grid and on a grid with 66 vertical levels, with the

Page 7, line 1 – LBC, [or lower boundary condition,]

Page 7, line 4 – resolutions

Page 7, line 4-5 – the different vertical sensitivities

Page 7, lines 9-10 – Use bold font for vectors and matrices.

Page 7, line 13 – kernel

Page 7, line 18 – add space before "being"

Page 7, line 19 – time series

Page 7, line 22 – with the AGAGE in situ time series

Page 7, line 23 – delete "here"

Page 7, line 25 – very well for

Page 7, line 27 – Unfiltered time series show similar results, with

Page 8, line 4 – Is this sentence intended to be a paragraph?

Page 8, line 5 – which is within

Page 8, line 6 – see Section 3.3)

Page 8, line 6 – BASCOE lower stratospheric HCFC-22 time series is slightly lower than these two

Page 8, line 8 – delete "of"

Page 8, line 11 – trend from the monthly mean lower

Page 8, line 19 – delete "as well"

Page 8, lines 20-21 – barely detectable in the MHD

Page 8, line 24 – height statistics

Page 8, line 24 – Could add the range of tropopause heights at JFJ over the year.

Page 8, line 30 – trend uncertainty

Page 8, line 35 – column trend

Page 8, line 38 – 1980s and early 1990s

Page 8, line 41 – 1980s

Page 9, lines 3-4 – calculated using AGAGE data

Page 9, line 8 – The 2005-2012 . . . are compared to the

Page 9, line 22 – (66 +- 6) %

Page 10, lines 1-2 – agreement for the

Page 10, line 9 – this improvement in retrieval strategy . . . essential for monitoring

Page 17, Figure 2, line 10 – The first eigenvector

Page 17, Figure 2, line 11 – instruments, and their second eigenvector has

Page 18, Figure 3, line 2 – time series of HCFC-22 total columns

Page 19, Figure 5 legend – Bruker Fourier fit

Page 19, Figure 6, line 8 – Seasonal cycle of HCFC-22 columns

Page 20, Table 1, lines 1 and 2 – "Homemade" or "homemade"? The latter is used elsewhere.

Page 20, Table 1, line 2 – text (Sect. 3.3)

---

## Author Comment (AC1) · 28 Jun 2019

**Improved FTIR retrieval strategy for HCFC-22 (CHClF2), comparisons with in situ and satellite datasets with the support of models, and determination of its long-term trend above Jungfraujoch**

Prignon et al., Atmos. Chem. Phys. Discuss., https://doi.org/10.5194/acp-2019-73, 2019

Authors' response to Anonymous Referee #1 (https://doi.org/10.5194/acp-2019-73-RC1)

We use blue text for referee's comments and black text for authors' response to these comments.

**General Comment**

This paper describes an improved HCFC-22 retrieval strategy from ground-based FTIR solar spectra at Jungfraujoch. They showed the possibility to distinguish the tropospheric and lower stratospheric partial columns from the FTIR spectra and compared their results with independent datasets (AGAGE and MIPAS) and models (BASCOE CTM and WACCM). However, there are some issues that should be clarified before this paper is published in ACP, which are described in the comments below.

> Thank you for your comments that will help to improve our manuscript. Here follows our response to your major and technical comments:

**Major Comments**

1) I have a concern on comparison between AGAGE (MHD and JFJ) data and FTIR mean tropospheric mixing ratio shown in Section 4.3. First of all, the way to calculate mean tropospheric mixing ratio from FTIR data is not described in detail. I think that SFIT-4 retrieval of FTIR spectra gives total column and vertical profiles with averaging kernel information. How the authors derive mean tropospheric mixing ratio from that information? Do they divide tropospheric HCFC-22 column between station altitude and 11.21 km by the amount of air molecule numbers at the same altitude range? Please explain in the text.

> We have modified the beginning of Section 4.3 (Section 4.4 in the modified manuscript) in order to precise how the tropospheric mean mixing ratio series is derived. The SFIT-4 algorithm returns, alongside total and partial columns, the vertical profile of the target species (i.e., HCFC-22) mixing ratios on the fixed 41-layer vertical grid used to model the atmosphere above our site. We then compute the mean mixing ratio for all the layers located between surface and 11.21 km altitude to get our "FTIR mean tropospheric mixing ratio".

2) Annual variations are seen in both derived total columns (Fig. 3) and tropospheric mean VMR (Fig. 4) in FTIR data, both having peaks in summer to fall. Such annual variations are not seen in AGAGE MHD nor JFJ data. However, there are no explanations nor discussion on the cause of the derived annual

variation. I wonder the derived annual variation may come from two reasons: a) The nature of FTIR measurement principle, i.e. measuring column amount above the observational station. The column amount might be affected by the height of tropopause height, which is higher in summer. b) The higher emission of HCFC-22 from the regional summertime use of air-conditioner, as is pointed out by Xiang et al. (2014). Please discuss more about the cause of the retrieved annual variation in FTIR data which are not seen it AGAGE data.

3) The scatter plot in Fig. 4 looks somewhat strange. We see many dots which are horizontally aligned. For example, there are several points for MHD value of ~145, but the next group jumps to >160. However, the actual trend of MHD values look more continuous. Please check if something wrong appeared or not to create this scatter plot.

4) In Section 4.4 (P.8, L.19), the authors claim that they do not show amplitude and phase of the seasonal cycle of tropospheric column series. However, as I mentioned in the above comment, differences in tropospheric annual variations are seen between FTIR retrieval and AGAGE data. I think they should show the figure which shows amplitude and phase of seasonal cycle of tropospheric columns as well, and discuss on the cause of such variation in more detail.

> Referee's comments 2, 3 and 4 are related and therefore addressed together in our response. The significant tropospheric annual cycle seen in our FITR time series is discussed in the original manuscript at the end of the section dealing with the lower stratospheric columns time series (Section 4.4), Page 8 Lines 18-24. Following your comment and those from reviewer #2, we realised that this discussion was not at the most appropriate place in the manuscript. Therefore, we have decided to switch Section 4.3 (troposphere) and Section 4.4 (stratosphere).

As explained in the information content Section (3.3), the main improvement resulting from this work is the determination of two independent times series from our spectra. Since the vertical limit between these two series is at 11.21 km (as indicated by the information content analysis), an altitude close to the mean tropopause height at Jungfraujoch, we assumed that the lower partial column or profile thus defined (from surface up to 11.21 km high) was mostly representative of the troposphere. Moreover, HCFC-22 seems to have no or a weak vertical gradient in the troposphere [see Figure 4 in Chirkov et al. (2016) or Figure 1 in Xiang et al. (2014)].

Nevertheless, the tropopause height varies throughout the year with minimum values in winter and maximum values in summer. Consequently, in winter, our mean mixing ratios should also include layers of low HCFC-22 concentration representative of the stratosphere, inducing this way a seasonal signal. Note that, for the period of comparison used in Section 4.3 (i.e., 1999-2018), the average tropopause height above Jungfraujoch is $(11.10 \pm 2.61)$ km [2σ; using the World Meteorological Organization (1957) thermal definition and the National Centers for Environmental Prediction pressure-temperature daily profiles]. We investigated the possible effect of the tropopause annual cycle by

computing tropospheric columns (below 11.21 km) from the HCFC-22 a priori profile used for the inversion (see Section 3.2) shifted up and down by about 2 km (i.e., a value representative of the spread around the mean tropopause height). This vertical shifting induces less than 2% of variation peak-to-peak on the tropospheric partial columns, i.e., less than the 7% peak-to-peak observed in our tropospheric column time series.

Another possible explanation to this tropospheric cycle is discussed by Chirkov et al. (2016). They compared mean upper tropospheric mixing ratios retrieved from MIPAS to surface in situ measurements and also noticed a significant annual cycle in their MIPAS time series, in contrast with the in situ data considered in their paper. They attributed this difference to the fact that their time series was capturing the intrusion of HCFC-22-poor stratospheric air at mid-latitudes Upper Troposphere / Lower Stratosphere (UTLS) at the time of the polar vortex breakdown (early spring).

Concerning now the higher emissions of HCFC-22 during summer from the regional summertime use of air-conditioner (Xiang et al., 2014), they should not be responsible of a strong annual cycle as depicted by our tropospheric time series. Indeed, these authors also stated that these emissions should be balanced by the increase of OH scavenging during summer. As written Page 8 Lines 18-24 of our original manuscript, the seasonal cycle in in situ series is in fact weak in amplitude, with broad maxima in winter and broad minima in summer.

Finally, the layered structure of the scatter plot of Figure 4 is actually caused by the seasonal cycle that is present in the FTIR mean tropospheric mixing ratio series, spreading the FTIR data around the in situ values (see Figure 1c hereafter).

Following the above discussion, we modified the end of the tropospheric Section (4.3) to more elaborate on the seasonal cycle retrieved in our data.

[Figure]

**Figure 1c: Daily coincidences between AGAGE Mace Head (MHD) and Jungfraujoch FTIR mean tropospheric VMR time series. Points are coloured according to the year of measurement (see right colour bar).**

**Minor Comments/Typos**

Figure 6: What is the value of the age of air? I think the right hand side axis to show the age of the air is missing.

> The age of air cycle is given in relative values (%). As written in Figure 6's caption, the mean age of air for the considered period is 2.96 year with a peak-to-peak amplitude range of 0.37 year.

P.1, L.33: The global warming potential of HCFC-22 should be 1810 (IPCC AR4) or 1780 (WMO O3 Assessment 2018), not 1760.

> We gave the value of AR5 (Table 8.A.1, Page 731 of IPCC, 2013)

**References**

Chirkov, M., Stiller, G. P., Laeng, A., Kellmann, S., von Clarmann, T., Boone, C. D., Elkins, J. W., Engel, A., Glatthor, N., Grabowski, U., Harth, C. M., Kiefer, M., Kolonjari, F., Krummel, P. B., Linden, A., Lunder, C. R., Miller, B. R., Montzka, S. A., Mühle, J., O'Doherty, S., Orphal, J., Prinn, R. G., Toon, G., Vollmer, M. K., Walker, K. A., Weiss, R. F., Wiegele, A. and Young, D.: Global HCFC-22 measurements with MIPAS: retrieval, validation, global distribution and its evolution over 2005–2012, Atmos. Chem. Phys., 16(5), 3345–3368, doi:10.5194/acp-16-3345-2016, 2016.

IPCC. Climate Change 2013: The Physical Science Basis. Contribution of Working Group I to the Fifth Assessment Report of the Intergovernmental Panel on Climate Change [Stocker, T.F., D. Qin, G.-K.

Plattner, M. Tignor, S.K. Allen, J. Boschung, A. Nauels, Y. Xia, V. Bex and P.M. Midgley (eds.)]. Cambridge University Press, Cambridge, United Kingdom and New York, NY, USA, 1535 pp, 2013

World Meteorological Organization (WMO). Definition of the tropopause, WMO Bull., 6, 136, 1957

Xiang, B., Patra, P. K., Montzka, S. A., Miller, S. M., Elkins, J. W., Moore, F. L., Atlas, E. L., Miller, B. R., Weiss, R. F., Prinn, R. G. and Wofsy, S. C.: Global emissions of refrigerants HCFC-22 and HFC-134a: Unforeseen seasonal contributions, Proc. Natl. Acad. Sci., 111(49), 17379–17384, doi:10.1073/pnas.1417372111, 2014.

---

## Author Comment (AC2) · 28 Jun 2019

**Improved FTIR retrieval strategy for HCFC-22 (CHClF2), comparisons with in situ and satellite datasets with the support of models, and determination of its long-term trend above Jungfraujoch**

Prignon et al., Atmos. Chem. Phys. Discuss., https://doi.org/10.5194/acp-2019-73, 2019

Authors' response to Anonymous Referee #2 (https://doi.org/10.5194/acp-2019-73-RC2)

We use blue text for referee's comments and black text for authors' response to these comments.

**General Comment**

The manuscript provides a clear and straight-forward description of the work. I recommend publication in ACP after the minor comments below are addressed.

> Thank you for your comments that help to improve this manuscript. Here follows our response to your specific and technical comments:

**Specific Comments**

Page 5, lines 36-39 – Is this the filtering referred to on page 7, line 26? Explain on page 7 the difference between the filtered and non-filtered in situ time series. This is the first time filtering is mentioned.

> Yes it is. These baseline data are directly provided by the AGAGE teams in charge of the in situ measurements using the method briefly explained in our manuscript and fully described in, e.g., O'Doherty et al. (2001) or Cunnold et al. (2002). In this work, we only compare our results with the AGAGE baseline data (i.e., filtered). The comparison between AGAGE Mace Head and AGAGE Jungfraujoch is done in order to emphasize that these series are in excellent agreement and that therefore we can also compare our results with Mace Head data. Finally, the difference between the AGAGE filtered and non-filtered series is shortly discussed in page 5 lines 36-39 of the original manuscript. The filtering is applied in order to produce time series representative of broad atmospheric regions, meaning that measurements directly influenced by regional pollution are filtered out.

Page 6, Section 4.1.2 – ACE-FTS HCFC-22 measurements are mentioned in the Introduction, so why aren't they included in the comparisons with the FTIR retrievals? Briefly explain why in this section. ACE data are also mentioned in the Data Availability section – is this because they was used in determining the systematic component of the Sa matrix?

> We decided to drop ACE-FTS HCFC-22 retrievals for the time series comparison because we wanted to keep an identical partial column height definition (i.e., 11.21-30 km as defined by the

information content in Section 3.3) between the datasets. Indeed, only about 20% of ACE-FTS measurements were remaining when selecting this altitude range. Nevertheless, following your comment, we decided to include ACE-FTS HCFC-22 data for the lower stratospheric decadal trend comparison. Section 5 and Table 4 have been thus modified to include the ACE-FTS 2005-2014 trends.

Page 6, lines 19-24 – Clarify what the lower boundary conditions are for – all trace gases in BASCOE? "only [a] few global observations are available . . ." – a few observations of what? Make clear which lower boundary conditions are derived from MLS and which from HGGC.

> Section 4.1.3 was modified following your comment in order to clarify what are the lower boundary condition and the simulation initial state.

Page 7, Section 4.3 – Some additional explanation should be provided regarding the comparison between the FTIR mean tropospheric mixing ratio and the in situ surface mixing ratio. Why compare the FTIR tropospheric mean mixing ratio rather than the FTIR surface value or lower tropospheric mean (I assume due to the information content)? How representative is the mean tropospheric mixing ratio of the surface mixing ratio? What error does this introduce? Figure 4 suggests that there are differences, since the FTIR mean values exhibit more seasonality than the in situ values.

> This comment is similar to the Major Comments of Referee #1. Firstly, we have modified the beginning of Section 4.3 in order to explain more clearly how the tropospheric mean mixing ratio series is build. As described in the information content and error budget Section (3.3), the characterization of the averaging kernel matrices showed that only two pieces of information could be extracted from the entire total column / profile retrieved by the inversion algorithm. The height of these partial columns / pieces of information are defined by the eigenvector of these averaging kernel matrices. Right hand side of Figure 2 further demonstrates that the separation between the two partial columns / pieces of information is located near 11 km, defining two broad ranges above and below that altitude, without any more vertical resolution. Therefore, it is not relevant nor correct to only take the mixing ratio retrieved for the surface layer.

Please refer to our response to Referee#1's comment for the discussion on the tropospheric cycle retrieved in our time series. As a short discussion was already in the original manuscript at the end of Section 4.4 (Comparison of lower stratospheric columns), we decided, for clarity, to switch Section 4.3 (tropospheric time series) and Section 4.4 and to more elaborate on the cause of the FTIR tropospheric cycle at the end of the tropospheric Section (i.e., 4.4 in the revised manuscript).

Page 8, lines 35-37 – Add these 1988-2017 total column trends to Tables 2-4.

> We added the 1988-2017 trends to Tables 2-4.

Page 10, lines 9-10 – Explain more explicitly how the improved retrieval strategy developed for HCFC-22 is transferable to other gases

> By this last sentence, we mean that we would be able to improve retrieval strategies of some other chlorine-bearing source gases (e.g., CFC-12) in order to retrieve partial columns, allowing then to independently characterize tropospheric and stratospheric trends. But for each case or species, a specific retrieval strategy has to be defined and optimized.

Page 18, Figure 4 – Why does the AGAGE vs. FTIR scatter plot have a layered structure, with a relatively constant AGAGE value for a range of FTIR values?

> As you pointed it out, FTIR tropospheric time series shows a significant annual cycle while in situ series do not. As a result, each year compared in the scatter plot are regrouped in "layer" (see Figure 1c hereafter).

[Figure]

**Figure 1c: Daily coincidences between AGAGE Mace Head (MHD) and Jungfraujoch FTIR mean tropospheric VMR time series. Points are coloured according their year of measurement (see right colour bar).**

Page 21-22, Tables 2-4 – Why are different time periods used for calculating the trends in the total columns, tropospheric columns, and lower stratospheric columns? This makes it difficult to directly compare them. The choices should be more clearly explained in the text.

> The periods used for calculating trends are selected in order to take the best from the compared time series but also to have consecutive trend periods of 10 years to highlight the changing trend values in the three last decades.

**Technical Corrections**

The manuscript should be reviewed carefully for grammatical and typographical errors. For example, there are many missing commas and hyphens, unnecessary or missing "s" on words, and other errors. Some are identified below, but this list is not exhaustive.

Thank you for all these technical corrections, the large majority of them have been included in the revised manuscript, here follows the exceptions:

Page 2, line 36 – not clear what Fohn refers to – delete?

> The authors referenced in the manuscript refer Föhn events to atmospheric situations as depressions over the Bay of Biscay bringing air into the Southern rim of the Alps and thus uplifting polluted air from surface to Jungfraujoch.

Page 6, line 17 – age-of-air study

> "age of air" is used in the literature.

**References**

Chirkov, M., Stiller, G. P., Laeng, A., Kellmann, S., von Clarmann, T., Boone, C. D., Elkins, J. W., Engel, A., Glatthor, N., Grabowski, U., Harth, C. M., Kiefer, M., Kolonjari, F., Krummel, P. B., Linden, A., Lunder, C. R., Miller, B. R., Montzka, S. A., Mühle, J., O'Doherty, S., Orphal, J., Prinn, R. G., Toon, G., Vollmer, M. K., Walker, K. A., Weiss, R. F., Wiegele, A. and Young, D.: Global HCFC-22 measurements with MIPAS: retrieval, validation, global distribution and its evolution over 2005–2012, Atmos. Chem. Phys., 16(5), 3345–3368, doi:10.5194/acp-16-3345-2016, 2016.

Cunnold, D. M., Steele, L. P., Fraser, P. J., Simmonds, P. G., Prinn, R. G., Weiss, R. F., Porter, L. W., O'Doherty, S., Langenfelds, R. L., Krummel, P. B., Wang, H. J., Emmons, L., Tie, X. X. and Dlugokencky, E.: In situ measurements of atmospheric methane at GAGE/AGAGE sites during 1985–2000 and resulting source inferences, J. Geophys. Res., 107(D14), 4225, doi:10.1029/2001JD001226, 2002.

O'Doherty, S., Simmonds, P. G., Cunnold, D. M., Wang, H. J., Sturrock, G. A., Fraser, P. J., Ryall, D., Derwent, R. G., Weiss, R. F., Salameh, P., Miller, B. R. and Prinn, R. G.: In situ chloroform measurements at Advanced Global Atmospheric Gases Experiment atmospheric research stations from 1994 to 1998, J. Geophys. Res. Atmos., 106(D17), 20429–20444, doi:10.1029/2000JD900792, 2001.

Xiang, B., Patra, P. K., Montzka, S. A., Miller, S. M., Elkins, J. W., Moore, F. L., Atlas, E. L., Miller, B. R., Weiss, R. F., Prinn, R. G. and Wofsy, S. C.: Global emissions of refrigerants HCFC-22 and HFC-134a: Unforeseen seasonal contributions, Proc. Natl. Acad. Sci., 111(49), 17379–17384, doi:10.1073/pnas.1417372111, 2014.

---

## Author Comment (AC3) · 28 Jun 2019

[revised manuscript text omitted]
[-1]) | 2.99 ± 0.05 (4.11 ± 0.07)% | 3.11 ± 0.19 (3.29 ± 0.2)% | 3.21 ± 0.11 (3.32 ± 0.11)% | 2.96 ± 0.39 (3.17 ± 0.42)% | 2.94 ± 0.11 (3.5 ± 0.13)% | 2.53 ± 0.16 (3.47 ± 0.22)% |

---

## Referee Report (RR1)

A 2$^{nd}$ Review of "Improved FTIR retrieval strategy for HCFC-22 (CHClF$_2$), comparisons with in situ and satellite datasets with the support of models, and determination of its long-term trend above Jungfraujoch" by M. Prignon et al.

<General Comments>

The authors made a good effort to revise the draft. I felt that most of my concerns are cleared now. However, I have the last concern on the content of the paper, which is described below. After responding my last concern with a minor revision of the draft, I think the paper is almost ready to be published in ACP.

<The Last Concern>

1) On the cause of annual variation in FTIR tropospheric column, the authors explained that it is attributed to the intrusion of HCFC-22-poor stratospheric air at mid-latitudes Upper Troposphere/Lower Stratosphere (UTLS) at the time of the polar vortex breakdown. Such upper tropospheric annual variation was also seen by MIPAS data (Chirkov et al., 2016). This explanation is convincing to some extent. However, here arises another question: Why such annual variation is not seen in AGAGE JFJ data? If the intrusion of HCFC-22-poor stratospheric air is the cause of the annual variation, it should also be captured in AGAGE in-situ data to some extent. It is hard to understand that the intrusion will not affect the free tropospheric high-altitude in-situ Jungfraujoch data at all. Is it because "non-polluted" AGAGE JFJ data is used for the plot? What will happen if "all" the AGAGE JFJ data are plotted? Please add some more discussion on the cause of this discrepancy in annual variation between FTIR tropospheric data and AGAGE in situ data in the draft.

---

## Author Response (AR2)

**Improved FTIR retrieval strategy for HCFC-22 (CHClF2), comparisons with in situ and satellite datasets with the support of models, and determination of its long-term trend above Jungfraujoch**

Prignon et al., Atmos. Chem. Phys. Discuss., https://doi.org/10.5194/acp-2019-73, 2019

Authors' response to Anonymous Referee #1's report (2nd review).

We use blue text for referee's comments and black text for authors' response to these comments.

**The Last Concern**

1) On the cause of annual variation in FTIR tropospheric column, the authors explained that it is attributed to the intrusion of HCFC 22 poor stratospheric air at mid latitudes Upper Troposphere/Lower Stratosphere (UTLS) at the time of the polar vortex breakdown. Such upper tropospheric annual variation was also seen by MIPAS data (Chirkov et al., 2016). This explanation is convincing to some extent. However, here arises another question: Why such annual variation is not seen in AGAGE JFJ data? If the intrusion of HCFC 22 poor stratospheric air is the cause of the annual variation, it should also be captured in AGAGE in situ data to some extent. It is hard to understand that the intrusion will not affect the free tropospheric high altitude in situ Jungfraujoch data at all Is it because non polluted AGAGE JFJ data is used for the plot? What will happen if all the AGAGE JFJ data are plotted? Please add some more discussion on the cause of this discrepancy in annual variation between FTIR tropospheric data and AGAGE in situ data in the draft.

> The two main questions are: 1) why such annual variation is not seen in AGAGE JFJ in situ data? 2) Does this difference is induced by the filtering of polluted events applied to the in situ time series?

The reviewer raised a good point in noting that it is surprising that the high altitude, and thus mostly free tropospheric, AGAGE JFJ surface station time series is not capturing the intrusion of HCFC-22-poor polar stratospheric air. We believe that there are several effects leading to this. Firstly, to answer to question 2), we can exclude the hypothesis that the difference in seasonality between FTIR and in situ time series is caused by the statistical filter applied on AGAGE time series. Figure AC1 indeed shows that there are only few and small pollution events (flagged in red) over the measurement time period. Over the almost 21000 measurements considered here (from August 2012 to March 2018), only 788 (~3.8%) are flagged as pollution event by the statistical algorithm. Note that the filter, aiming at building a time series representative of background conditions (i.e., identifying high HCFC-22 concentration values), does not filter out the lower values.

As to question 1), it is likely that the lower stratospheric and Upper Troposphere/Lower Stratosphere (UTLS) cycle as discussed by Chirkov et al. (2016) and Nevison et al. (2004) and observed in our retrievals is dampened towards the surface. Indeed, not so many stratospheric intrusions should reach the surface at JFJ, and if so, their magnitude should be reduced by mixing with other air masses. Regarding the depleting events observed in the HCFC-22 surface time series, they could indeed correspond to stratospheric intrusions, but they more likely correspond to advection of low-HCFC-22 air masses from the Atlantic Ocean (Uglietti et al., 2011) and generally from lower latitude regions with reduced HCFC-22 abundances (see the gradient in HCFC-22 from southern to northern latitudes in Figure AC2). Another point that gives us confidence is the fact that the Zeppelin (Ny-Ålesund, 78.9°N) in situ time series is neither capturing the effect of the polar vortex breakdown during the late winter/early spring (on filtered and non-filtered data). In conclusion that is to say that HCFC-22 surface air concentration in situ measurements performed at the mostly free tropospheric JFJ station are unlikely to be representative of the UTLS and thus capture stratospheric polar air mass intrusions.

Furthermore, we would also like to stress that the difference in the seasonal cycles at Jungfraujoch could also be artificially amplified by the non-constant vertical sensitivity of our FTIR retrievals. Total averaging kernels (see left frame of manuscript's Figure 2 and Figure AC3 below) indeed show a peak of sensitivity in the UTLS, where the cycle is dominant compared to lower altitude annual signals (see lower left frame of Figure 15 in Chirkov et al., 2016).

In conclusion, we believe we would need further elements or measurements (e.g., high-resolution vertical profiles of HCFC-22 in the troposphere) to investigate the contrasted seasonal modulations observed in the troposphere and to identify the main factors driving them. Nevertheless, we showed that our FTIR mean tropospheric time series compares very well with AGAGE in situ measurements from Mace Head and JFJ in terms of absolute values and trends over the last two decades. Further discussions on the surface HCFC-22 seasonality at JFJ will be left for future work.

> Following this discussion we added in Page 8 Line 33 the discussion about the effect of the vertical sensitivity of our retrievals on the retrieved tropospheric seasonal cycle.

[Figure]

**Figure AC1: HCFC-22 in situ measurements performed at the Jungfraujoch AGAGE station. See original manuscript for filtering method.**

[Figure]

**Figure AC2: HCFC-22 surface mole fraction (ppt) from seven AGAGE stations. Downloaded from http://agage.eas.gatech.edu/data_archive/data_figures/monthly/pdf/HCFC-22_mm.pdf (2019-08-19)**

[Figure]

**Figure AC3: Total layer averaging kernels derived from measurement of the year 2015. See manuscript for method.**

**References**

[revised manuscript text omitted]
 | $8.13 \pm 0.08$ $(3.75 \pm 0.04)\%$ | $8.52 \pm 0.57$ $(5.88 \pm 0.39)\%$ | $7.09 \pm 0.37$ $(3.41 \pm 0.18)\%$ | $8.6 \pm 0.28$ $(3.03 \pm 0.1)\%$ | $7.98 \pm 0.29$ $(2.57 \pm 0.09)\%$ |
| BASCOE | | $7.21 \pm 0.1$ $(5.3 \pm 0.07)\%$ | $\underline{6.94 \pm 0.15}$ $\underline{(3.57 \pm 0.08)\%}$ | $\underline{8.98 \pm 0.2}$ $(3.35 \pm 0.08)\%$ | – |
| WACCM | | $6.63 \pm 0.13$ $(5.16 \pm 0.1)\%$ | $6.3 \pm 0.12$ $\underline{(3.41 \pm 0.07)\%}$ | $\underline{8.53 \pm 0.2}$ $(3.36 \pm 0.08)\%$ | – |

**Table 3: HCFC-22 tropospheric trends over JFJ. The uncertainties are given for the 2σ level following the Santer et al. (2000) approach. Trends are underlined when not significantly different from FTIR trends. Relative trends (% yr⁻¹) are given with respect to the yearly mean of the middle year of the considered period. [a] Time frame enlarged in order to encompass the 2012 low quality measurements period (see 3.3).**

| Tropospheric trends (ppt/year) | 1988 – 2017 ($10^{+13}$ molec cm⁻²) | 1999 - 2008 | 2008 - 2017 | 2012-2017 |
|---|---|---|---|---|
| FTIR Mean Tropospheric mixing ratio | $5.1 \pm 0.06$ $(3.41 \pm 0.07)\%$ | $6.54 \pm 0.35$ $(3.72 \pm 0.2)\%$ | $5.31 \pm 0.42$ $(2.29 \pm 0.18)\%$ | $4.04 \pm 0.73$ $(1.72 \pm 0.31)\%$ (2011-2017)[a] |
| AGAGE Mace Head | | $\underline{6.39 \pm 0.1}$ $\underline{(3.66 \pm 0.06)\%}$ | $\underline{5.36 \pm 0.12}$ $\underline{(2.27 \pm 0.05)\%}$ | $\underline{4.2 \pm 0.08}$ $\underline{(1.71 \pm 0.03)\%}$ |
| AGAGE JFJ | | - | - | $\underline{4.05 \pm 0.12}$ $\underline{(1.65 \pm 0.04)\%}$ |

**Table 4. HCFC-22 lower stratospheric trends over JFJ. The uncertainties are given for the 2σ level following the Santer et al. (2000) approach. Relative trends (% yr$^{-1}$) are given with respect to the yearly mean of the middle year of the considered period. Trends are underlined when not significantly different from the FTIR trends.**

| | FTIR JFJ | FTIR JFJ | MIPAS | ACE-FTS | BASCOE CTM | WACCM |
|---|---|---|---|---|---|---|
| | 1988-2017 | 2005-2014 | 2005-2012 | 2005-2014 | 2005-2014 | 2005-2014 |
| Lower stratospheric trends ($10^{+13}$molec cm$^{-2}$ yr$^{-1}$) | 2.99 ± 0.05 (4.11 ± 0.07)% | 3.11 ± 0.19 (3.29 ± 0.2)% | 3.21 ± 0.11 (3.32 ± 0.11)% | 2.96 ± 0.39 (3.17 ± 0.42)% | 2.94 ± 0.11 (3.5 ± 0.13)% | 2.53 ± 0.16 (3.47 ± 0.22)% |

---

## Author Response (AR3)

**Improved FTIR retrieval strategy for HCFC-22 (CHClF2), comparisons with in situ and satellite datasets with the support of models, and determination of its long-term trend above Jungfraujoch**

Prignon et al., Atmos. Chem. Phys. Discuss., https://doi.org/10.5194/acp-2019-73, 2019

Authors' response to Anonymous Referee #1 (https://doi.org/10.5194/acp-2019-73-RC1)

We use blue text for referee's comments and black text for authors' response to these comments.

**General Comment**

This paper describes an improved HCFC-22 retrieval strategy from ground-based FTIR solar spectra at Jungfraujoch. They showed the possibility to distinguish the tropospheric and lower stratospheric partial columns from the FTIR spectra and compared their results with independent datasets (AGAGE and MIPAS) and models (BASCOE CTM and WACCM). However, there are some issues that should be clarified before this paper is published in ACP, which are described in the comments below.

> Thank you for your comments that will help to improve our manuscript. Here follows our response to your major and technical comments:

**Major Comments**

1) I have a concern on comparison between AGAGE (MHD and JFJ) data and FTIR mean tropospheric mixing ratio shown in Section 4.3. First of all, the way to calculate mean tropospheric mixing ratio from FTIR data is not described in detail. I think that SFIT-4 retrieval of FTIR spectra gives total column and vertical profiles with averaging kernel information. How the authors derive mean tropospheric mixing ratio from that information? Do they divide tropospheric HCFC-22 column between station altitude and 11.21 km by the amount of air molecule numbers at the same altitude range? Please explain in the text.

> We have modified the beginning of Section 4.3 (Section 4.4 in the modified manuscript) in order to precise how the tropospheric mean mixing ratio series is derived. The SFIT-4 algorithm returns, alongside total and partial columns, the vertical profile of the target species (i.e., HCFC-22) mixing ratios on the fixed 41-layer vertical grid used to model the atmosphere above our site. We then compute the mean mixing ratio for all the layers located between surface and 11.21 km altitude to get our "FTIR mean tropospheric mixing ratio".

2) Annual variations are seen in both derived total columns (Fig. 3) and tropospheric mean VMR (Fig. 4) in FTIR data, both having peaks in summer to fall. Such annual variations are not seen in AGAGE MHD nor JFJ data. However, there are no explanations nor discussion on the cause of the derived annual

variation. I wonder the derived annual variation may come from two reasons: a) The nature of FTIR measurement principle, i.e. measuring column amount above the observational station. The column amount might be affected by the height of tropopause height, which is higher in summer. b) The higher emission of HCFC-22 from the regional summertime use of air-conditioner, as is pointed out by Xiang et al. (2014). Please discuss more about the cause of the retrieved annual variation in FTIR data which are not seen it AGAGE data.

3) The scatter plot in Fig. 4 looks somewhat strange. We see many dots which are horizontally aligned. For example, there are several points for MHD value of ~145, but the next group jumps to >160. However, the actual trend of MHD values look more continuous. Please check if something wrong appeared or not to create this scatter plot.

4) In Section 4.4 (P.8, L.19), the authors claim that they do not show amplitude and phase of the seasonal cycle of tropospheric column series. However, as I mentioned in the above comment, differences in tropospheric annual variations are seen between FTIR retrieval and AGAGE data. I think they should show the figure which shows amplitude and phase of seasonal cycle of tropospheric columns as well, and discuss on the cause of such variation in more detail.

> Referee's comments 2, 3 and 4 are related and therefore addressed together in our response. The significant tropospheric annual cycle seen in our FITR time series is discussed in the original manuscript at the end of the section dealing with the lower stratospheric columns time series (Section 4.4), Page 8 Lines 18-24. Following your comment and those from reviewer #2, we realised that this discussion was not at the most appropriate place in the manuscript. Therefore, we have decided to switch Section 4.3 (troposphere) and Section 4.4 (stratosphere).

As explained in the information content Section (3.3), the main improvement resulting from this work is the determination of two independent times series from our spectra. Since the vertical limit between these two series is at 11.21 km (as indicated by the information content analysis), an altitude close to the mean tropopause height at Jungfraujoch, we assumed that the lower partial column or profile thus defined (from surface up to 11.21 km high) was mostly representative of the troposphere. Moreover, HCFC-22 seems to have no or a weak vertical gradient in the troposphere [see Figure 4 in Chirkov et al. (2016) or Figure 1 in Xiang et al. (2014)].

Nevertheless, the tropopause height varies throughout the year with minimum values in winter and maximum values in summer. Consequently, in winter, our mean mixing ratios should also include layers of low HCFC-22 concentration representative of the stratosphere, inducing this way a seasonal signal. Note that, for the period of comparison used in Section 4.3 (i.e., 1999-2018), the average tropopause height above Jungfraujoch is $(11.10 \pm 2.61)$ km [$2\sigma$; using the World Meteorological Organization (1957) thermal definition and the National Centers for Environmental Prediction pressure-temperature daily profiles]. We investigated the possible effect of the tropopause annual cycle by

computing tropospheric columns (below 11.21 km) from the HCFC-22 a priori profile used for the inversion (see Section 3.2) shifted up and down by about 2 km (i.e., a value representative of the spread around the mean tropopause height). This vertical shifting induces less than 2% of variation peak-to-peak on the tropospheric partial columns, i.e., less than the 7% peak-to-peak observed in our tropospheric column time series.

Another possible explanation to this tropospheric cycle is discussed by Chirkov et al. (2016). They compared mean upper tropospheric mixing ratios retrieved from MIPAS to surface in situ measurements and also noticed a significant annual cycle in their MIPAS time series, in contrast with the in situ data considered in their paper. They attributed this difference to the fact that their time series was capturing the intrusion of HCFC-22-poor stratospheric air at mid-latitudes Upper Troposphere / Lower Stratosphere (UTLS) at the time of the polar vortex breakdown (early spring).

Concerning now the higher emissions of HCFC-22 during summer from the regional summertime use of air-conditioner (Xiang et al., 2014), they should not be responsible of a strong annual cycle as depicted by our tropospheric time series. Indeed, these authors also stated that these emissions should be balanced by the increase of OH scavenging during summer. As written Page 8 Lines 18-24 of our original manuscript, the seasonal cycle in in situ series is in fact weak in amplitude, with broad maxima in winter and broad minima in summer.

Finally, the layered structure of the scatter plot of Figure 4 is actually caused by the seasonal cycle that is present in the FTIR mean tropospheric mixing ratio series, spreading the FTIR data around the in situ values (see Figure 1c hereafter).

Following the above discussion, we modified the end of the tropospheric Section (4.3) to more elaborate on the seasonal cycle retrieved in our data.

[Figure]

**Figure 1c: Daily coincidences between AGAGE Mace Head (MHD) and Jungfraujoch FTIR mean tropospheric VMR time series. Points are coloured according to the year of measurement (see right colour bar).**

**Minor Comments/Typos**

Figure 6: What is the value of the age of air? I think the right hand side axis to show the age of the air is missing.

> The age of air cycle is given in relative values (%). As written in Figure 6's caption, the mean age of air for the considered period is 2.96 year with a peak-to-peak amplitude range of 0.37 year.

P.1, L.33: The global warming potential of HCFC-22 should be 1810 (IPCC AR4) or 1780 (WMO O3 Assessment 2018), not 1760.

> We gave the value of AR5 (Table 8.A.1, Page 731 of IPCC, 2013)

Page 6, lines 19-24 – Clarify what the lower boundary conditions are for – all trace gases in BASCOE? "only [a] few global observations are available . . ." – a few observations of what? Make clear which lower boundary conditions are derived from MLS and which from HGGC.

> Section 4.1.3 was modified following your comment in order to clarify what are the lower boundary condition and the simulation initial state.

Page 7, Section 4.3 – Some additional explanation should be provided regarding the comparison between the FTIR mean tropospheric mixing ratio and the in situ surface mixing ratio. Why compare the FTIR tropospheric mean mixing ratio rather than the FTIR surface value or lower tropospheric mean (I assume due to the information content)? How representative is the mean tropospheric mixing ratio of the surface mixing ratio? What error does this introduce? Figure 4 suggests that there are differences, since the FTIR mean values exhibit more seasonality than the in situ values.

> This comment is similar to the Major Comments of Referee #1. Firstly, we have modified the beginning of Section 4.3 in order to explain more clearly how the tropospheric mean mixing ratio series is build. As described in the information content and error budget Section (3.3), the characterization of the averaging kernel matrices showed that only two pieces of information could be extracted from the entire total column / profile retrieved by the inversion algorithm. The height of these partial columns / pieces of information are defined by the eigenvector of these averaging kernel matrices. Right hand side of Figure 2 further demonstrates that the separation between the two partial columns / pieces of information is located near 11 km, defining two broad ranges above and below that altitude, without any more vertical resolution. Therefore, it is not relevant nor correct to only take the mixing ratio retrieved for the surface layer.

Please refer to our response to Referee#1's comment for the discussion on the tropospheric cycle retrieved in our time series. As a short discussion was already in the original manuscript at the end of Section 4.4 (Comparison of lower stratospheric columns), we decided, for clarity, to switch Section 4.3 (tropospheric time series) and Section 4.4 and to more elaborate on the cause of the FTIR tropospheric cycle at the end of the tropospheric Section (i.e., 4.4 in the revised manuscript).

Page 8, lines 35-37 – Add these 1988-2017 total column trends to Tables 2-4.

> We added the 1988-2017 trends to Tables 2-4.

Page 10, lines 9-10 – Explain more explicitly how the improved retrieval strategy developed for HCFC-22 is transferable to other gases

> By this last sentence, we mean that we would be able to improve retrieval strategies of some other chlorine-bearing source gases (e.g., CFC-12) in order to retrieve partial columns, allowing then to independently characterize tropospheric and stratospheric trends. But for each case or species, a specific retrieval strategy has to be defined and optimized.

Page 18, Figure 4 – Why does the AGAGE vs. FTIR scatter plot have a layered structure, with a relatively constant AGAGE value for a range of FTIR values?

> As you pointed it out, FTIR tropospheric time series shows a significant annual cycle while in situ series do not. As a result, each year compared in the scatter plot are regrouped in "layer" (see Figure 1c hereafter).

[Figure]

**Figure 1c: Daily coincidences between AGAGE Mace Head (MHD) and Jungfraujoch FTIR mean tropospheric VMR time series. Points are coloured according their year of measurement (see right colour bar).**

Page 21-22, Tables 2-4 – Why are different time periods used for calculating the trends in the total columns, tropospheric columns, and lower stratospheric columns? This makes it difficult to directly compare them. The choices should be more clearly explained in the text.

> The periods used for calculating trends are selected in order to take the best from the compared time series but also to have consecutive trend periods of 10 years to highlight the changing trend values in the three last decades.

**Technical Corrections**

The manuscript should be reviewed carefully for grammatical and typographical errors. For example, there are many missing commas and hyphens, unnecessary or missing "s" on words, and other errors. Some are identified below, but this list is not exhaustive.

Thank you for all these technical corrections, the large majority of them have been included in the revised manuscript, here follows the exceptions:

Page 2, line 36 – not clear what Fohn refers to – delete?

> The authors referenced in the manuscript refer Föhn events to atmospheric situations as depressions over the Bay of Biscay bringing air into the Southern rim of the Alps and thus uplifting polluted air from surface to Jungfraujoch.

Page 6, line 17 – age-of-air study

> "age of air" is used in the literature.

> The two main questions are: 1) why such annual variation is not seen in AGAGE JFJ in situ data? 2) Does this difference is induced by the filtering of polluted events applied to the in situ time series?

The reviewer raised a good point in noting that it is surprising that the high altitude, and thus mostly free tropospheric, AGAGE JFJ surface station time series is not capturing the intrusion of HCFC-22-poor polar stratospheric air. We believe that there are several effects leading to this. Firstly, to answer to question 2), we can exclude the hypothesis that the difference in seasonality between FTIR and in situ time series is caused by the statistical filter applied on AGAGE time series. Figure AC1 indeed shows that there are only few and small pollution events (flagged in red) over the measurement time period. Over the almost 21000 measurements considered here (from August 2012 to March 2018), only 788 (~3.8%) are flagged as pollution event by the statistical algorithm. Note that the filter, aiming at building a time series representative of background conditions (i.e., identifying high HCFC-22 concentration values), does not filter out the lower values.

As to question 1), it is likely that the lower stratospheric and Upper Troposphere/Lower Stratosphere (UTLS) cycle as discussed by Chirkov et al. (2016) and Nevison et al. (2004) and observed in our retrievals is dampened towards the surface. Indeed, not so many stratospheric intrusions should reach the surface at JFJ, and if so, their magnitude should be reduced by mixing with other air masses. Regarding the depleting events observed in the HCFC-22 surface time series, they could indeed correspond to stratospheric intrusions, but they more likely correspond to advection of low-HCFC-22 air masses from the Atlantic Ocean (Uglietti et al., 2011) and generally from lower latitude regions with reduced HCFC-22 abundances (see the gradient in HCFC-22 from southern to northern latitudes in Figure AC2). Another point that gives us confidence is the fact that the Zeppelin (Ny-Ålesund, 78.9°N) in situ time series is neither capturing the effect of the polar vortex breakdown during the late winter/early spring (on filtered and non-filtered data). In conclusion that is to say that HCFC-22 surface air concentration in situ measurements performed at the mostly free tropospheric JFJ station are unlikely to be representative of the UTLS and thus capture stratospheric polar air mass intrusions.

Furthermore, we would also like to stress that the difference in the seasonal cycles at Jungfraujoch could also be artificially amplified by the non-constant vertical sensitivity of our FTIR retrievals. Total averaging kernels (see left frame of manuscript's Figure 2 and Figure AC3 below) indeed show a peak of sensitivity in the UTLS, where the cycle is dominant compared to lower altitude annual signals (see lower left frame of Figure 15 in Chirkov et al., 2016).

In conclusion, we believe we would need further elements or measurements (e.g., high-resolution vertical profiles of HCFC-22 in the troposphere) to investigate the contrasted seasonal modulations observed in the troposphere and to identify the main factors driving them. Nevertheless, we showed that our FTIR mean tropospheric time series compares very well with AGAGE in situ measurements from Mace Head and JFJ in terms of absolute values and trends over the last two decades. Further discussions on the surface HCFC-22 seasonality at JFJ will be left for future work.

> Following this discussion we added in Page 8 Line 33 the discussion about the effect of the vertical sensitivity of our retrievals on the retrieved tropospheric seasonal cycle.

[Figure]

**Figure AC1: HCFC-22 in situ measurements performed at the Jungfraujoch AGAGE station. See original manuscript for filtering method.**

[Figure]

**Figure AC2: HCFC-22 surface mole fraction (ppt) from seven AGAGE stations. Downloaded from http://agage.eas.gatech.edu/data_archive/data_figures/monthly/pdf/HCFC-22_mm.pdf (2019-08-19)**

[Figure]

**Figure AC3: Total layer averaging kernels derived from measurement of the year 2015. See manuscript for method.**

**References**

[revised manuscript text omitted]
⁻¹) | $2.99 \pm 0.05$ $(4.11 \pm 0.07)\%$ | $3.11 \pm 0.19$ $(3.29 \pm 0.2)\%$ | $3.21 \pm 0.11$ $(3.32 \pm 0.11)\%$ | $2.96 \pm 0.39$ $(3.17 \pm 0.42)\%$ | $2.94 \pm 0.11$ $(3.5 \pm 0.13)\%$ | $2.53 \pm 0.16$ $(3.47 \pm 0.22)\%$ |